



# FALL3D-8.0: a computational model for atmospheric transport and deposition of particles, aerosols and radionuclides. Part I: model physics and numerics

Arnau Folch[1], Leonardo Mingari[1], Natalia Gutierrez[1], Mauricio Hanzich[1], Giovanni Macedonio[2], and Antonio Costa[3]

[1]CASE Department, Barcelona Supercomputing Center, Barcelona, Spain
[2]Istituto Nazionale di Geofisica e Vulcanologia, Osservatorio Vesuviano, Napoli, Italy
[3]Istituto Nazionale di Geofisica e Vulcanologia, Sezione di Bologna, Bologna, Italy

**Correspondence:** Arnau Folch (afolch@bsc.es)

**Abstract.** This manuscript presents *FALL3D-8.0*, the last version release of an open-source code with 15+ years of track record and a growing number of users in the volcanological and atmospheric communities. The code has been redesigned and rewritten from scratch in the framework of the EU Center of Excellence for Exascale in Solid Earth (*ChEESE*) in order to overcome legacy issues and allow for successive optimisations that are already planned in the preparation of the code towards extreme-scale computing. However, this baseline version already contains substantial improvements in terms of model physics, solving algorithms, and code accuracy and performance. The code, originally conceived for atmospheric dispersal and deposition of tephra particles, has been extended to model other types of particles, aerosols and radionuclides. The solving strategy has also been changed, replacing the former central-differences scheme for a high-resolution central-upwind scheme derived from finite volumes, which minimises numerical diffusion even in presence of sharp concentration gradients and discontinuities. The parallelisation strategy, Input/Output (I/O), model pre-process workflows and memory management have also been reconsidered, leading to substantial improvements on code scalability, efficiency, and overall capability to handle much larger problems. This paper details the *FALL3D-8.0* model physics and the numerical implementation of the code.

## 1 Introduction

*FALL3D* is an open-source off-line Eulerian model for atmospheric passive transport and deposition based on the so-called Advection-Diffusion-Sedimentation (ADS) equation. The model, originally developed for inert volcanic particles (tephra), has a track record of 50+ publications on different model applications and code validation, as well as an ever-growing community of users worldwide, including academia, private, research, and several institutions tasked with operational forecast of volcanic ash clouds. The first versions of *FALL3D* (v1.x series) appeared back in 2003 (Costa and Macedonio, 2004), at that time the code being serial and written in FORTRAN-77. Code improvements at different levels have been continuously incorporated since then, including relevant milestones leading to code version upgrades, *e.g.* the coupling with 1D Buoyant Plume Theory (BPT) models to define eruption column sources (versions v2.x, 2004), the introduction of the Lax-Wendorff (LW) central





differences scheme for solving the ADS equation (v3.x, 2005) and other algorithmic improvements (Costa et al., 2006), full code rewriting in FORTRAN-90 and distributed memory parallelisation by means of Message Passing Interface (MPI) (v5.x, 2007), first implementations of operational workflows to forecast ash cloud dispersal and fallout (Folch et al., 2008, 2009), and

several other improvements (e.g. de la Cruz et al., 2016) until v7.3.4 release in 2018.

Along these 15+ years, *FALL3D* has been used in multiple applications (e.g. Folch, 2012) including, among others, assessment of hazard from tephra fallout at different volcanoes (e.g. Scaini et al., 2012; Selva et al., 2014; Sandri et al., 2016), impacts of ash cloud dispersal on civil aviation (e.g. Sulpizio et al., 2012; Bonasia et al., 2013; Biass et al., 2014; Scaini et al., 2014), obtaining relevant eruption source parameters (e.g. Folch et al., 2012; Parra et al., 2016; Poret et al., 2017), impacts of past

super-eruptions on climate, environment, and humans (e.g. Costa et al., 2012, 2014; Martí et al., 2016), operational forecast of ash clouds and tephra fallout (e.g. Bear-Crozier et al., 2012; Collini et al., 2013; Poulidis et al., 2019), modelling of ash resuspension events (e.g. Folch et al., 2014; Mingari et al., 2017), or model validation (e.g. Scollo et al., 2010; Corradini et al., 2011; Osores et al., 2013). However, as occurs in other long-lived community codes, code legacy limitations have appeared with time on, *e.g.*, lack of code performance and poor scalability on hundreds/thousands of cores, constrains on portability and

adaption to emerging hardware architectures, difficulties for code refactoring that is needed in order to widen the spectrum of model applications, etc. With time, the proper address of these aspects required of substantial code refactoring or even code rewriting, a substantially time-consuming task in terms of human effort. This has recently been possible in the frame of the European Center of Excellence for Exascale in Solid Earth (*ChEESE*), which includes *FALL3D* as one of its flagship codes.

This paper describes *FALL3D-8.0*, a new model version upgrade in which the code has been completely written from scratch,

mostly in FORTRAN-90 but mixed with some FORTRAN-2003 functionalities. In addition to dramatic improvements on different levels (extended model physics and applications, numerical algorithmic and code performance), v8.0 provides also with a baseline that will allow the incorporation of developments and optimisations. However, the heterogeneity of model users has been considered when rewriting the code, which can still run on platforms spanning from a laptop to a large supercomputer.

This manuscript starts first outlining what's new in *FALL3D-8.0* (Sec. 2) with respect to the previous code release (v7.3.4)

and then describes the physical model and related governing equations and parameterisations (Sec. 3). The next section focuses on the numerical implementation (Sec. 4), including coordinate mappings and scaling, spatial discretisation, and a new solving strategy based on the Kurganov-Tadmor scheme (Kurganov and Tadmor, 2000) that can be combined either with a fourth-order Runge-Kutta or with a first-order Euler to integrate explicitly in time. These two solver options allow users to choose, respectively, between better solver accuracy or higher computational efficiency. In any case, it will be shown how

the Kurganov-Tadmor finite-volume-like formulation is much less diffusive than the previous scheme implemented in v7.x, an important feature when one aims at modelling substances encompassing sharp gradients of concentration. Sections 5 and 6 focus, respectively, on the (pre-process) model workflow and on the new code paralellisation strategy and related memory optimisations, including a comparison on model performance and scalability with respect to v7.x. This paper shows only one illustrative example of *FALL3D-8.0* model results for ash dispersal from the 2011 Cordón Caulle eruption. A companion paper

(Prata et al., 2019) contains the second part, including a detailed *FALL3D-8.0* model validation for several simulations that are





part of the new benchmark suite of the code. Finally, Section 7 wraps the conclusions of the manuscript and outlines which will be the next steps for further model optimisation.

## 2 New features in v8.0

*FALL3D-8.0* introduces substantial improvements at different levels. From the point of view of model physics:

- The code has been generalised to deal with species different from tephra, including other kinds of particles, aerosols and radionuclides. Different types of species have been defined and can be simulated using independent sets of bins.

- Weibull and bi-Weibull particle Total Grain Size Distributions (TGSDs) can now be generated in addition to Lognormal distributions. On the other hand, *FALL3D-8.0* can estimate TGSDs of tephra particles directly from magma viscosity (magma composition) and eruption column height following the fit proposed by Costa et al. (2016a).

- An "effective bin" flag has been introduced. For a given specie, only those bins having this flag "on" are actually simulated, whereas bins tagged as "off" are frozen in terms of transport and used only for source term characterisation. This option has been added because several model parameterisations for the emission (source) term depend on the whole granulometric spectrum of particles but, at the same time, model users are often interested only on a subset of the particle spectra (*e.g.* fine volcanic ash for long-range tephra dispersal simulations).

- For the specie "tephra", several classes of particle aggregates can now be defined in certain aggregation options, differently than the single-class aggregation option available in v7.x.

- Model parameterisations for physics have been revised to include more recent studies. Meteorological drivers have also been updated to add recent datasets (*e.g.* ERA-5) and to remove deprecated options.

- Periodic boundary conditions are now possible, permitting simulations on domains spanning from local to semi-global
  (pole singularities still remain).

- For some species, the initial model condition can be furnished from satellite retrievals. This "data insertion" option is a preliminary step towards a full model data assimilation using ensembles.

From the point of view of numerics and code performance:

- The solving strategy has been changed to a high-resolution central-upwind scheme (Kurganov and Tadmor, 2000), op-
  tionally combined with a 4th-order Runge-Kutta explicit scheme. This replaces the former Lax-Wendorff (LW) central-difference scheme, which was known to be over-diffusive in case of sharp concentration gradients or discontinuities.

- New coordinate mappings have been added. These include a new vertical $\sigma$-coordinate system with linear decay that smooths low-level numerical oscillations over complex terrains.



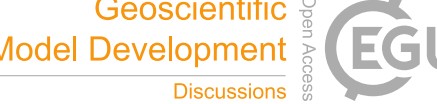

- A new parallelisation strategy exists based on a full 3D domain decomposition. The former trivial parallelisation on particle bins in v7.x has been removed because, in case of interaction among bins, it yielded to unnecessary communication penalties, resulting on poor code scalability.

- A much more efficient memory management exists to exploit contiguous cache memory positions along each spatial dimension. In some model configurations, this can imply a substantial gain on computing time.

- Parallel model I/O using netCDF-Par and parallel model pre-process. In addition, all the pre-process auxiliary programs have been embedded within the main code (a single multipurpose executable exists in v8.0). The code can now be run to perform different tasks individually or sequentially through a single parallel workflow. As a result, all the pre-process, modelling and post-process workflows can now run as a single execution concatenating all tasks and without needing to write/read intermediate files to/from disk. In large problems, this saves substantial disk space and I/O time because the required model input data (*e.g.* interpolated meteorological fields) are already stored in each processor memory when running the *FALL3D* model task.

- A hierarchy of MPI communicators has been defined. This is actually not active yet in v8.0 but gives flexibility to extend some code functionalities in a near future with little refactoring effort. For example, plans for future versions include ensemble modelling using the Parallel Data Assimilation Framework (PDAF) or model nesting. These will require of teams of processors associated to different ensemble members or model grids respectively.

## 3  Physical model

### 3.1  Model governing equations

In continuum mechanics, the general form describing the passive transport of a substance mixed within a fluid (air) in a domain $\Omega$ derives from mass conservation which, in conservative flux-form and using the Eulerian specification, reads:

$$\frac{\partial c}{\partial t} + \nabla \cdot \mathbf{F} + \nabla \cdot \mathbf{G} + \nabla \cdot \mathbf{H} = S - I \qquad \text{in} \qquad \Omega \tag{1}$$

where $\mathbf{F} = c\,\mathbf{u}$ is the advective flux, $\mathbf{G} = c\,\mathbf{u_s}$ is the sedimentation flux, $\mathbf{H} = -\mathbb{K}\nabla c$ is the diffusive flux, and $S$ and $I$ are the source and sink terms respectively. In the expressions above, $t$ denotes time, $c$ is concentration (in $\mathrm{kg\,m^{-3}}$), $\mathbf{u}$ is the fluid velocity vector (*i.e.* wind velocity), $\mathbf{u_s}$ is the terminal settling velocity of the substance, and $\mathbb{K}$ is the diffusion tensor (in $\mathrm{m^2 s^{-1}}$). Note that the definition of the diffusive flux $\mathbf{H}$ explicitly assumes the Fick's first law of diffusion.

Boundary conditions are imposed at the Dirichlet $\Gamma_D$ (inflow), Neumann $\Gamma_N$ (outflow), and Robin $\Gamma_R$ (ground) parts of the boundary of the computational domain $\Gamma$ (with $\Gamma = \Gamma_D \cup \Gamma_N \cup \Gamma_R$ and $\Gamma_D \cap \Gamma_N \cap \Gamma_R = 0$) as:

$$\begin{cases} c = \bar{c} & \text{at } \Gamma_D; \\ \mathbf{n} \cdot \mathbf{H} = 0 & \text{at } \Gamma_N; \\ \mathbf{n} \cdot (\mathbf{H} + \mathbf{G}) = \mathbf{n} \cdot \mathbf{D} & \text{at } \Gamma_R \end{cases} \tag{2}$$





where $\bar{c}$ is the concentration prescribed at inflow (typically $\bar{c} = 0$), $\mathbf{n}$ is the outwards unit normal vector, and $\mathbf{D} = c\,\mathbf{u_d}$ is the ground deposition flux ($\mathbf{u_d}$ is the ground deposition velocity). Note that when the deposition flux $\mathbf{D}$ coincides with the sedimentation flux $\mathbf{G}$ (*i.e.* when $\mathbf{u_s} = \mathbf{u_d}$), the boundary condition at ground reduces to the standard free flow condition

imposed at $\Gamma_N$.

Equation (1) is the so-called Advection-Diffusion-Sedimentation (ADS) equation and, in *FALL3D-8.0*, has been extended to handle passive transport of other substances different from tephra. In a general sense, substances in *FALL3D-8.0* are grouped in 3 broad categories: particles, aerosols, and radionuclides. The category "particles" includes any inert substance characterised by a sedimentation velocity. The category "aerosol" refers in *FALL3D-8.0* to substances potentially non-inert (*i.e.* with chemical

or phase changes mechanisms) and having a negligible sedimentation velocity. Finally, the category "radionuclides" refers to isotope particles subject to radioactive decay. Each of these categories admits, in turn, different sub-categories or "species", defined internally as structures of data that inherit the parent category properties (see Table 1). For example, particles can be subdivided into tephra or mineral dust; aerosol species can include $H_2O$, $SO_2$, etc.; and radionuclides can include several isotope species. Finally, each sub-category of species is tagged with an attribute name that is used for descriptive purposes

only.

Depending on the specie(s) under consideration, the mass source $S$ and sink $I$ terms in (1) can be decomposed as:

$$S = S^e + S^a + S^r + S^c$$
$$I = I^w + I^a + I^r + I^c \tag{3}$$

where the superscripts denote emission source terms ($S^e$; see Sec. 3.2.3), wet deposition sinks ($I^w$; see Sec. 3.2.4), particle

aggregation source and sinks ($S^a$ and $I^a$ respectively; see Sec. 3.2.6), radioactive decay source and sinks ($S^r$ and $I^r$ respectively; see Sec. 3.2.7), and chemical reactions source and sinks ($S^c$ and $I^c$). Note that *FALL3D-8.0* does not account for aerosol chemistry yet. However, the code has been designed to allow incorporating this functionality in future versions in a straightforward manner.

When the ADS equation (1) is discretised, species in the mixture are binned in $n_b$ discrete "classes" or bins, so that the total

concentration $c$ at any point of the domain decomposes as the sum of bin concentrations, *i.e.*, $c = \sum_{i=1}^{n_b} c_i$. Substitution of bin discretisation in (1) yields to $n_b$ equations (one per discrete bin), each formally identical to (1):

$$\frac{\partial c_i}{\partial t} + \nabla \cdot \mathbf{F}_i + \nabla \cdot \mathbf{G}_i + \nabla \cdot \mathbf{H}_i = S_i - I_i \qquad i = 1 : n_b \tag{4}$$

Note that the $n_b$ equations for bins can be coupled by means of the source and sink terms, which define the eventual transfer of mass among different bins, *e.g.* in case of particle aggregation/disegregation, chemical reactions, formation of child radionu-

clides, etc.





## 3.2 Model parameterisations

Parameterisations in *FALL3D* have been revised and updated, removing deprecated options and adding new options available from more recent studies. Table 2 shows the parameterisations implemented in *FALL3D-8.0* that described in the following
subsections.

### 3.2.1 Diffusion tensor

The atmospheric flow is characterised by large horizontal to vertical aspect ratios of wind velocities and length scales, as well as by an anisotropic momentum diffusion with the horizontal diffusion coefficient being typically much larger than the vertical one (e.g. Schaefer-Rolffs and Becker, 2013). For this reason, model diffusion that accounts for sub-grid scale atmospheric eddies
is typically assumed anisotropic, with two distinct eddy diffusion coefficients along the horizontal and vertical dimensions:

$$\mathbb{K} = \begin{pmatrix} k_h & 0 & 0 \\ 0 & k_h & 0 \\ 0 & 0 & k_v \end{pmatrix} \tag{5}$$

In the case of *FALL3D-8.0*, the horizontal coefficient $k_h$ can be either assumed constant or parameterised as in Byun and Schere (2006):

$$\frac{1}{k_h} = \frac{1}{k_{ht}} + \frac{1}{k_{hn}} \tag{6}$$

where:

$$k_{ht} = 2\,\alpha^2 \Delta_g^2 \sqrt{S_\Gamma^2 + S_\Lambda^2} \tag{7}$$
$$= \alpha^2 \Delta_g^2 \sqrt{\left(\frac{\partial u}{\partial x} - \frac{\partial v}{\partial y}\right)^2 + \left(\frac{\partial u}{\partial x} + \frac{\partial v}{\partial y}\right)^2}$$

$$k_{hn} = k_{ref}\left(\frac{\Delta_{ref}}{\Delta_g}\right) \tag{8}$$

In the equations above, $\Delta_g$ is a characteristic grid cell measure, $\alpha \cong 0.28$ denotes the Smagorinsky constant, $S_\Gamma$ and $S_\Lambda$ are the stretching and shear strength (*i.e.* the two components of the bi-dimensional wind deformation), and $k_{ref}$ is a reference horizontal diffusion for a reference grid cell size $\Delta_{ref}$ (*FALL3D-8.0* considers $k_{ref} = 8000$ m$^2$ s$^{-1}$m for $\Delta_{ref} = 4$ km). Equation (6) was proposed by Byun and Schere (2006) to overcome the dependency of horizontal diffusion on grid resolution, and combines a Smagorinsky sub-grid scale (SGS) model giving the diffusion by transport ($k_{ht}$) with a formula that counteracts
numerical over-diffusion in coarse discretisations ($k_{hn}$) so that the smaller between $k_{ht}$ and $k_{hn}$ dominates. In this way, the effect of the transportive diffusion is minimised for coarse grids, whereas for fine discretisations the numerical diffusion term is reduced.





The vertical diffusion coefficient $k_v$ can also be either assumed constant or parameterised according to the similarity theory and distinguishing among surface layer, Atmospheric Boundary Layer (ABL), and free atmosphere (e.g. Neale et al., 2010):

$$
\quad k_v = \begin{cases} \dfrac{\kappa z u_*}{\phi_h} & \text{for} \quad z << h_p \\[2ex] \dfrac{\kappa z u_*}{\phi_h}\left(1-\dfrac{z}{h}\right)^2 & \text{for} \quad z < h_p \\[2ex] l_c^2 \left|\dfrac{\partial u_h}{\partial z}\right| F_c(Ri) & \text{for} \quad z > h_p \end{cases} \tag{9}
$$

where $\kappa$ is the von Karman constant ($\kappa = 0.4$), $z$ is the distance from the ground, $u_*$ is wind friction velocity, $\phi_h$ is the atmospheric stability function for temperature, $h_p$ is the ABL height, $l_c$ is a characteristic length scale, $u_h$ is the horizontal wind velocity modulus, and $F_c$ is a stability function which depends on the Richardson number $Ri$. For $l_c$ and $F_c$, *FALL3D-8.0* adopts the relationships used by the CAM-4.0 model (Neale et al., 2010):

$$
\quad l_c = \left(\frac{1}{\kappa z} + \frac{1}{\lambda_c}\right)^{-1} \tag{10}
$$

$$
F_c(Ri) = \begin{cases} \dfrac{1}{1+10Ri(1+8Ri)} & \text{stable} \quad (Ri > 0) \\[2ex] \sqrt{1-18Ri} & \text{unstable} \quad (Ri < 0) \end{cases} \tag{11}
$$

where $\lambda_c$ is the so-called asymptotic length scale ($\lambda_c \approx 30$m). The atmospheric stability function for temperature $\phi_h$ is calculated as:

$$
\quad \phi_h = \begin{cases} \beta_h + \dfrac{z}{L} & z/L > 1 \quad \text{stable atmosphere} \\[2ex] 1 + \beta_h \dfrac{z}{L} & 0 \le z/L \le 1 \quad \text{nearly neutral} \\[2ex] \left(1-\gamma_h \dfrac{z}{L}\right)^{-1/2} & z/L < 0 \quad \text{unstable atmosphere} \end{cases} \tag{12}
$$

where $\beta_h = 5$, $\gamma_h = 15$, and $L$ is the Monin-Obukhov length, defined as:

$$
L = \frac{u_*^2 \bar{\theta}_v}{\kappa g \theta_*} \tag{13}
$$

with $g$ denoting gravity, $\bar{\theta}_v$ the mean potential virtual temperature, and $\theta_*$ the potential temperature scale. The parameters in (13) (*i.e.* $L$ or $u_*/\theta_*$) are ideally furnished by the driving meteorological model. If not and alternatively, *FALL3D-8.0* estimates
the friction velocity $u_*$ and the potential temperature scale $\theta_*$ from the potential virtual temperature $\theta_v$ and the Richardson bulk number $Ri_b$ as (Louis, 1979; Jacobson, 1999):

$$
Ri_b = \frac{g\left[\theta_v(z_r) - \theta_v(z_o)\right](z_r - z_o)}{\bar{\theta}_v u_h(z_r)^2} \tag{14}
$$





where $z_r$ and $z_o$ denote the reference and the ground roughness heights, and $\bar{\theta}_v$ the average between the two vertical levels. Given $Ri_b$, one can estimate $u_*$ and $\theta_*$ as:

$$u_* \approx \frac{\kappa u_h(z_r)}{\ln(z_r/z_o)} \sqrt{G_m(Ri_b)} \tag{15}$$

$$\theta_* \approx \frac{\kappa^2 \, u_h \left[\theta_v(z_r) - \theta_v(z_o)\right]}{u_* Pr_t \ln^2(z_r/z_o)} \sqrt{G_h(Ri_b)} \tag{16}$$

where $Pr_t$ is the turbulent Prandtl number ($Pr_t \approx 1$) and the stability functions $G_m$ and $G_h$ are given by (Louis, 1979; Jacobson, 1999):

$$Gm = \begin{cases} 1 - \dfrac{9.4 Ri_b}{1 + 70\kappa^2(|Ri_b| \, z_r/z_o)^{0.5}/\ln^2(z_r/z_o)} & Ri_b \leq 0 \\[3mm] \dfrac{1}{(1 + 4.7 Ri_b)^2} & Ri_b > 0 \end{cases} \tag{17}$$

$$Gh = \begin{cases} 1 - \dfrac{9.4 Ri_b}{1 + 50\kappa^2(|Ri_b| \, z_r/z_o)^{0.5}/\ln^2(z_r/z_o)} & Ri_b \leq 0 \\[3mm] \dfrac{1}{(1 + 4.7 Ri_b)^2} & Ri_b > 0 \end{cases} \tag{18}$$

### 3.2.2 Sedimentation velocity

Particle bins in the model are assumed to settle down with a sedimentation velocity $\mathbf{u_s} = (0, 0, -w_s)$ equal to its terminal velocity:

$$w_s = \sqrt{\frac{4g \, (\rho_p - \rho_a) \, d}{3 \, C_d \rho_a}} \tag{19}$$

where $\rho_a$ and $\rho_p$ denote air and particle density, $d$ is the particle equivalent diameter, and $C_d$ is the drag coefficient that depends on the Reynolds number, $Re = du_s/\nu_a$ ($\nu_a = \mu_a/\rho_a$ being the kinematic viscosity of air and $\mu_a$ its dynamic viscosity). For irregular particles, the drag coefficient $C_d$ has to be obtained from experimental measurements. *FALL3D-8.0* includes several parameterisations derived from laboratory results using natural and synthetic particles and that cover a wide range of particle sizes and shapes (characterised by sphericity, by circularity, or by some other model shape factor). Model options for the drag coefficient $C_d$ include:

1. The *GANSER* model (Ganser, 1993):

$$\begin{aligned} C_d = &\frac{24}{Re K_1} \left\{ 1 + 0.1118 \left(Re \, K_1 K_2\right)^{0.6567} \right\} \\ &+ \frac{0.4305 K_2}{1 + \dfrac{3305}{Re \, K_1 K_2}} \end{aligned} \tag{20}$$





where $K_1 = 3/[(d_n/d) + 2\Psi^{-0.5}]$ and $K_2 = 10^{1.8148(-\mathrm{Log}\Psi)^{0.5743}}$ are two shape factors, $d_n$ is the average between the minimum and the maximum axis, and $\Psi$ is the particle sphericity ($\Psi = 1$ for spheres). For calculating the sphericity, is practical to use the concepts of "operational" and "working sphericity", $\Psi_{work}$ introduced by Wadell (1933) and Aschenbrenner (1956), which are based on the determination of the volume and of the three dimensions of a particle respectively:

$$\Psi_{work} = 12.8 \frac{(P^2 Q)^{1/3}}{1 + P(1+Q) + 6\sqrt{1 + P^2(1+Q^2)}} \tag{21}$$

with $P = S/I$, $Q = I/L$, where $L$ is the longest particle dimension, $I$ is the longest dimension perpendicular to $L$, and $S$ is the dimension perpendicular to both $L$ and $I$.

2. The *PFEIFFER* model (Pfeiffer et al., 2005), based on the interpolation of previous relationships by Walker et al. (1971) and Wilson and Huang (1979):

$$C_d = \begin{cases} \dfrac{24}{Re}\varphi^{-0.828} + 2\sqrt{1-\varphi} & Re \leq 10^2 \\ 1 - \dfrac{1 - C_d|_{Re=10^2}}{900}(10^3 - Re) & 10^2 \leq Re \leq 10^3 \\ 1 & Re \geq 10^3 \end{cases} \tag{22}$$

where $\varphi = (b+c)/2a$ is the particle aspect ratio ($a \geq b \geq c$ denote the particle semi-axes).

3. The *DIOGUARDI* model (Dioguardi et al., 2018):

$$\begin{aligned} C_d = & \frac{24}{Re}\left(\frac{1-\xi}{Re} + 1\right)^{0.25} \\ & + \frac{24}{Re}(0.1806 Re^{0.6459})\xi^{-Re^{0.08}}\frac{0.4251}{1 + \frac{6880.95}{Re^2}\xi^{5.05}} \end{aligned} \tag{23}$$

where $\xi$ is a particle shape factor (sphericity to circularity ratio), for which Dioguardi et al. (2018) suggested an empirical correlation with sphericity $\Psi$ as $\xi = 0.83\Psi$.

Note that, in any case, the terminal velocity $w_s$ is defined by a triplet $(d, \rho_p, \Psi)$. As a result, particles with similar values of the three parameters can be grouped within the same model bin.

### 3.2.3 Emissions

The emission source term for the $i$-th bin ($S_i^e$ term in the bin equations (4)) gives the mass per unit of time and volume (units of $\mathrm{kg\,m^{-3}\,s^{-1}}$) released at each point (cell) of the computational domain. *FALL3D-8.0* can generate and handle multiple types of emission sources, internally defined as a data structure made of $n_p$ discrete points, each "tagged" with a time-varying position and bin emission rate (source strength) $M_{ip}$ (in $\mathrm{kg\,s^{-1}}$). As a result, $S_i^e$ in a model grid cell results from summing emissions from all point sources laying within the cell volume $V$:

$$S_i^e = \sum_{p=1}^{n_p} M_{ip}/V \tag{24}$$



The total source strength $M_o$ results from summing over all source points and bins, *i.e.*:

$$M_o = \sum_{i=1}^{n_b} \sum_{p=1}^{n_p} M_{ip} = \sum_{i=1}^{n_b} M_i \tag{25}$$

Table 3 summarises the different (exclusive) options available in *FALL3D-8.0* for the emission term and related source strength. In detail:

1. The *POINT* option assumes that all mass is emitted from a single point ($n_p = 1$) located at height $z_t$ above ground level:

$$M_i = \begin{cases} f_i M_o & z = z_t \\ 0 & z \neq z_t \end{cases} \tag{26}$$

where $f_i$ is the $i$-th bin mass fraction.

2. The *HAT* option defines a uniform vertical line of $n_p$ source points spanning in height from $z_b$ (bottom) to $z_t$ (top) above the ground (*i.e.* with thickness $z_t - z_b$):

$$M_i = \begin{cases} \dfrac{f_i \, M_o}{n_p} & z_b \leq z \leq z_t \\ 0 & \text{otherwise} \end{cases} \tag{27}$$

Note that this option includes as end-members the *POINT* option (if $z_b = z_t$) and a vertically uniform emission from ground to top (if $z_b = 0$).

3. The *SUZUKI* option (Suzuki, 1983; Pfeiffer et al., 2005) assumes a mushroom-like vertical distribution of $n_p$ emission points depending on two dimensionless parameters $A$ and $\lambda$:

$$M_i = \frac{f_i \, M_o}{n_p} \left[ \left( 1 - \frac{z}{z_t} \right) e^{A\left(\frac{z}{z_t} - 1\right)} \right]^{\lambda} \quad 0 \leq z \leq z_t \tag{28}$$

The Suzuki parameter $A$ controls the vertical location of the maximum of the emission profile, whereas the parameter $\lambda$ controls the distribution of the emitted mass around the maximum.

When any of the previous source options is defined for volcanic plumes, it is useful to prescribe the total source strength (eruption mass flow rate) $M_o$ in terms of the eruption column height $H$ because this parameter is easier to be obtained from direct observations. To this purpose, *FALL3D-8.0* includes two relationships that correlate $M_o$ with $H$ based on empirical observations and on 1D plume model simulations respectively. The first and simplest case considers the fit proposed by Mastin et al. (2009):

$$M_o = aH^{4.15} \tag{29}$$

where $a = 140.8$ is a constant and $H$ is the eruption column height expressed in $\mathrm{km}$ above the eruptive vent. Alternatively, the 1D model fit by Woodhouse et al. (2016) can also be used to provide $M_o$ depending on the surrounding





atmospheric conditions:

$$M_o = c_o N^3 H^4 f(W) \tag{30}$$

where $N$ is the Brunt-Väisälä frequency, $c_o$ is a constant, and $f$ is a function of the parameter $W = 1.44\dot{\gamma}/N$ given by:

$$f(W) = \left(\frac{1 + bW + cW^2}{1 + aW}\right)^4 \tag{31}$$

with the coefficients $a = 0.87 + 0.05\beta/\alpha$, $b = 1.09 + 0.32\beta/\alpha$, and $c = 0.06 + 0.03\beta/\alpha$, being $\alpha$ and $\beta$ the radial and cross-wind plume entrainment coefficients (Costa et al., 2016b) and $\dot{\gamma}$ the mean shear rate of wind. The constant $c_o$ in (30) depends on the conditions at the vent:

$$c_o = \frac{0.35\,\alpha^2\,\rho_a C_a T_a}{g\left[(C_v n_o + C_s(1 - n_o))T_o - C_a T_a\right]} \tag{32}$$

being $C_s$, $C_v$ and $C_a$ the specific heat capacities at constant pressure of the solid pyroclasts, gas phase, and air respectively, $T_a$ and $T_o$ the air and vent magma mixture temperatures, and $n_0$ the mass gas fraction.

4. The *PLUME* option, valid only for volcanic plumes, uses the *FPLUME-1.0* model (Folch et al., 2016) embedded in *FALL3D-8.0*. *FPLUME-1.0* is a steady-state 1D cross-section-averaged eruption column model based on the buoyant plume theory (BPT). The model accounts for plume bending by wind, entrainment of ambient moisture, effects of water phase changes, particle fallout and re-entrainment, and a model for wet aggregation of ash particles in the presence of liquid water or ice. As opposed to the previous cases, the *PLUME* source option automatically determines a bin-dependent vertical distribution of mass and computes height from $M_o$ or vice-versa by solving an inverse problem.

5. The *RESUSPENSION* option considers the remobilisation and resuspension by wind of soil particles (*e.g.* mineral dust or volcanic ash previously deposited on the ground). Up to 3 different emission schemes are available in *FALL3D-8.0* to obtain the vertical flux of suspended particles, from which $M_o$ is obtained by multiplying by the associated surface cell area (see Folch et al., 2014, for details). Tipically, the emission schemes for mineral dust are formulated in terms of the friction velocity. For example, emission scheme 1 (Westphal et al., 1987) considers:

$$F_V = \begin{cases} 0 & u_* < u_{*t} \\ 10^{-5} u_*^4 & u_* \geq u_{*t} \end{cases} \tag{33}$$

where $F_V$ is the vertical flux (in $\mathrm{kg\,m^{-2}\,s^{-1}}$), occurring only above a (constant) threshold friction velocity $u_{*t}$ ($u_*$ given in $\mathrm{ms^{-1}}$). An important limitation of (33) is that the vertical flux does not depend neither on particle size nor soil moisture. However, despite its simplicity, this parameterisation can be useful when information on soil characteristics (*e.g.* particle sizes and densities, moisture, roughness, etc.) is unavailable or poorly constrained. Emission scheme 2 (Marticorena and Bergametti, 1995; Marticorena et al., 1997) considers:

$$F_V = \begin{cases} 0 & u_* < u_{*t} \\ S_c \dfrac{\rho_a}{g} u_*^3 \left(1 - \dfrac{u_{*t}^2}{u_*^2}\right)\left(1 + \dfrac{u_{*t}}{u_*}\right) & u_* \geq u_{*t} \end{cases} \tag{34}$$





where the experimental coefficient $S_c$ (in $\mathrm{cm}^{-1}$) depends on the amount of available fine particles in the soil, and the threshold friction velocity is given by:

$$u_{*t} = \begin{cases} \dfrac{0.129K}{(1.928Re^{0.092} - 1)^{0.5}} & 0.03 < Re \le 10 \\ 0.129K(1 - 0.0858e^{-0.0617(Re-10)}) & Re > 10 \end{cases} \tag{35}$$

with $K = \sqrt{\dfrac{\rho_p g d}{\rho_a}\left(1 + \dfrac{0.006}{\rho_p g d^{2.5}}\right)}$ and $Re = 1331 \times d^{1.56}$ (the lower bound of the fit corresponds to particles of $\approx 10\mu\mathrm{m}$ in size). Note that in (35), $\rho_p$ and $\rho_a$ are particle and air densities (expressed in $\mathrm{g/cm^3}$), $g$ is gravity (in $\mathrm{cm/s^2}$), $d$ is the particle size (in cm), $Re$ is the Reynolds number parameterised as a function of the particle size, and $u_{*t}$ is given in cm/s.

Finally, emission scheme 3 (Shao et al., 1993; Shao and Leslie, 1997; Shao and Lu, 2000) considers that the uplift from surface of the fine fraction of soil particles is controlled by the bombardment of saltating particles of larger sizes ($\ge 63\ \mu m$), which breaks the cohesive forces of smaller particles. Based on theoretical and experimental results, Shao et al. (1993) found an expression for the vertical flux of dust particles of size $d$ ejected by the impact of saltating particles of size $d_s$:

$$F_V(d, d_s) = \frac{\alpha(d, d_s)}{u_{*t}^2(d)} F_H(d_s) \tag{36}$$

where $\alpha$ (in $\mathrm{m\,s^{-2}}$) is the coefficient of sandblasting efficiency determined experimentally (Shao and Leslie, 1997) and $F_H$ is the horizontal flux (in $\mathrm{kg\,m^{-1}\,s^{-1}}$) of saltating particles:

$$F_H(d_s) = \begin{cases} 0 & u_* < u_{*t}(d_s) \\ c_o \dfrac{\rho_a u_*^3}{g}\left(1 - \dfrac{u_{*t}^2(d_s)}{u_*^2}\right) & u_* \ge u_{*t}(d_s) \end{cases} \tag{37}$$

where $c_o$ is an empirical dimensionless constant close to 1. In this scheme, the threshold friction velocity $u_{*t}(d)$ is given by:

$$u_{*ts} = \sqrt{0.0123\left(\frac{\rho_p g d}{\rho_a} + \frac{\gamma}{\rho_a d}\right)} \tag{38}$$

where $\gamma$ is an experimental parameter ranging between $1.65 \times 10^{-4}$ and $5 \times 10^{-4}\ \mathrm{kg\,s^2}$ (a value of $3 \times 10^{-4}\ \mathrm{kg\,s^2}$ is assumed in *FALL3D-8.0*).

### 3.2.4 Deposition mechanisms

In *FALL3D-8.0*, dry and wet deposition mechanisms can be activated for any type of bin below a certain particle/aerosol size. Dry deposition on the ground is imposed prescribing the deposition velocity through a Robin boundary condition in (2). *FALL3D-8.0* admits two dry deposition parameterisations, which describe the vertical depositional fluxes by Brownian diffusion and inertial impaction, parameterised through the Schmidt and the Stokes number respectively. The first option considers





the mass-consistent formulation proposed by Venkatram and Pleim (1999):

$$u_d = w_s + \frac{w_s}{1 - e^{-(r_a + r_b)w_s}} \approx w_s + \frac{1}{r_a + r_b} \tag{39}$$

where $r_a$ describes the effects of aerodynamic resistance and $r_b$ the quasi-laminar resistance (e.g. Brandt et al., 2002, and references therein). The aerodynamic resistance $r_a$ can be calculated as:

$$r_a = \frac{1}{ku_*}\left[\ln\left(\frac{z}{z_o}\right) - \phi_h\left(\frac{z}{L}\right)\right] \tag{40}$$

with $z_o$ denoting the ground roughness height and $\phi_h$ the atmospheric stability function for temperature given by (12). The quasi-laminar resistance $r_b$ can be expressed in terms of the Schmidt number $Sc = \nu/D$ and Stokes number $St = w_s u_*^2/(g\nu)$ (with $\nu$ kinematic viscosity of air, and $D$ molecular diffusivity of particles) (e.g. Brandt et al., 2002):

$$r_b = \frac{1}{u_*\left(Sc^{-2/3} + 10^{-3/St}\right)} \tag{41}$$

The second option is that proposed by Feng (2008), which essentially differs from (39) in the estimation of $r_b$:

$$u_d = w_s + \frac{1}{r_a + 1/(u_* c_1 e^{-0.5[(Re^* - c_2)/c_3]^2} + a u_*^b)} \tag{42}$$

where $c_1 = 0.0226$, $c_2 = 40300$ and $c_3 = 15330$ are dimensionless parameterisation constants, $Re^*$ is the Reynolds number (computed with the friction velocity $u_*$), and $a$ and $b$ are coefficients that depend on the particle size and surface characteristics. Note that Feng (2008) gives $a$ and $b$ best-fit values for 7 land use categories and 4 aerosol size modes: nuclei (up to 0.1 μm), accumulation (up to 2.5 μm), coarse (up to 10 μm), and giant (up to 100 μm). A cut-off is assumed above this size because the sedimentation velocity term $w_s$ dominates and therefore the dry deposition contribution can be neglected.

Wet deposition mechanisms in *FALL3D-8.0* are assumed to occur only within the Atmospheric Boundary Layer (ABL) and the corresponding sink term in (3) is parameterised as:

$$I^w = \Lambda c \tag{43}$$

where $\Lambda$ differs for in-cloud (ic) and below-cloud (bc) sinks. For below-cloud scavenging (precipitation), $\Lambda_{bc}$ is estimated from the total precipitation rate as (e.g. Brandt et al., 2002; Jung and Shao, 2006):

$$\Lambda_{bc} = a P^b \tag{44}$$

where $P$ is the precipitation rate (in $\mathrm{mm\,h^{-1}}$), and $a = 8.4 \times 10^{-5}$ and $b = 0.79$ are two empirical constants. For in-cloud scavenging (rainout), the model considers a parameterisation based on the atmospheric relative humidity $RH$ (in %) as in Brandt et al. (2002):

$$\Lambda_{ic} = \begin{cases} 0 & \text{for } RH < RH_t \\ A_{RH}\dfrac{RH - RH_t}{RH_s - RH_t} & \text{for } RH \geq RH_t \end{cases} \tag{45}$$

with $A_{RH} = 3.5 \times 10^{-5}$, $RH_t = 80\%$ (threshold value), and $RH_s = 100\%$ (saturation value). Two critical particle cut-off sizes of 100 and 1 μm are assumed for below and in-cloud scavenging respectively.



### 3.2.5 Gravity spreading of the umbrella region

Large explosive volcanic eruptions can generate gravity-driven transport mechanisms that dominate over passive transport close to the vent and cause a radial spreading of the cloud (e.g. Woods and Kienle, 1994; Sparks et al., 1997). In order to simulate this mechanism, *FALL3D-8.0* includes a gravity current model (see Costa et al., 2013, and the Erratum published in June 2019). This option consists on adding a radial velocity field to the background wind, so that contributions from both passive and density-driven mechanisms are accounted for. The added radial wind is centred above the eruptive vent in the umbrella region, and extended up to a radius $R$ given by:

$$R = \left(\frac{3\lambda Nq}{2\pi}\right)^{1/3} t^{2/3} \tag{46}$$

where $t$ is time since eruption onset, $\lambda$ is an empirical constant constrained to $\approx 0.2$ from Direct Numerical Simulations (Suzuki and Koyaguchi, 2009), $N$ is the Brunt-Väisälä frequency, and $q$ is the volumetric flow rate into the umbrella region, estimated as (Morton et al., 1956; Suzuki and Koyaguchi, 2009; Costa et al., 2013, as correct in Erratum 2019):

$$q = \frac{c\,k^{1/2}M_o^{3/4}}{N^{5/4}} \tag{47}$$

where $M_o$ is the total source strength (*i.e.* mass eruption rate), $k$ is the air entrainment coefficient, and $c$ a constant that from varies from tropical to mid-latitude/polar locations:

$$c = \begin{cases} 0.43 & \mathrm{m^3\,kg^{-3/4}\,s^{-3/2}} \text{ for tropical} \\ 0.87 & \mathrm{m^3\,kg^{-3/4}\,s^{-3/2}} \text{ mid-latitude/polar} \end{cases} \tag{48}$$

Given the radius $R$, the radial velocity field as a function of distance $r$ is calculated as (Costa et al., 2013):

$$u_r(r) = \frac{3}{4} u_r(R) \frac{R}{r}\left(1 + \frac{1}{3}\frac{r^2}{R^2}\right) \qquad (0 \le r \le R) \tag{49}$$

where $u_r(R)$ is the front velocity:

$$u_r(R) = \left(\frac{2\lambda Nq}{3\pi}\right)^{1/2} \frac{1}{\sqrt{R}} \tag{50}$$

In order to avoid sudden jumps at the gravity current front, *FALL3D-8.0* interpolates the front velocity $u_r(R)$ with far field wind velocity using an exponential decay function of the cloud thickness $h$ as:

$$\exp[-d/(4h)] \tag{51}$$

where $d$ is the distance from current front.

### 3.2.6 Aggregation

Aggregation of tephra particles can occur inside the eruptive columns or even downwind in ash clouds during atmospheric dispersion, thereby affecting the sedimentation dynamics and deposition of volcanic ash. *FALL3D-8.0* includes some simple





a-priori aggregation options and a wet aggregation model (Costa et al., 2010; Folch et al., 2010) that can be activated for tephra bins. The a-priori options consist on user-defined or empirically-based pre-defined fractions of aggregating classes being transferred to one or more class of aggregates at the source points (*i.e.* aggregation is performed before transport). In contrast, in the wet aggregation model, ash particles aggregate on a single effective class of diameter $d_A$, *i.e.* aggregation

only affects tephra bins with diameter smaller than $d_A$, typically in the range 100-300 μm. This option can run embedded in *FPLUME-1.0* or as stand-alone. Consider a tephra grain size distribution in which $k$ particle bins can aggregate. Then, the aggregation model defines the source ($S^a$) and sink ($I^a$) bin terms for the corresponding $k + 1$ bins as:

$$\begin{cases} S^a_{k+1} = \sum_{j=1}^{k} I^a_j \\ I^a_j = \dfrac{\pi d_j^3 \rho_j}{6} \dot{n}_j \quad j = 1 : k \end{cases} \tag{52}$$

where $d_j (< d_A)$ and $\rho_j$ are, respectively, the diameter and density of particles in bin $j$, and $\dot{n}_j$ is the number of particles per

380 unit volume and time that aggregate. The model assumes that this is proportional to the total particle decay per unit volume $\dot{n}_{tot}$, *i.e.*:

$$\dot{n}_j \approx \frac{N_j}{\sum_{i=1}^{k} N_i} \dot{n}_{tot} \tag{53}$$

where

$$N_j = k_f \left( \frac{d_A}{d_j} \right)^{d_f} \tag{54}$$

is the number of primary particles of diameter $d_j$ in an aggregate of diameter $d_A$, $k_f$ is a fractal pre-factor ($k_f \approx 1$), and $D_f$ is the fractal exponent ($D_f \leq 3$). The model estimates the total particle decay per unit time $\dot{n}_{tot}$ integrating the coagulation kernel over all particle sizes, depending on the sticking efficiency times a collision frequency function which accounts for Brownian motion, collision due to turbulence as a result of inertial effects, laminar and turbulent fluid shear, and differential sedimentation (see Costa et al., 2010; Folch et al., 2016, for details).

### 3.2.7 Radioactive decay

*FALL3D-8.0* can handle the fate of radioactive material dispersed from accidental releases (e.g., Brandt et al., 2002; Leelössya et al., 2018). Five common species of radionuclides have been implemented, Cesium $^{134}$Cs, $^{137}$Cs and Iodine $^{131}$I, which decay to stable isotopes, and $^{90}$Sr, which decays to the unstable isotope $^{90}$Y (see Table 4).

Radionuclide species need to specify the source ($S^r_n$) and the sink ($I^r_n$) terms in (3), associated to the radioactive production

or decay of the isotope $n$ respectively. The radioactive decay term indicates the mass per unit volume of the isotopes of type $n$ that decays per unit time:

$$I^r_n = k_r \, c_n \tag{55}$$

where $c_n$ is concentration (expressed in $\text{kg m}^{-3}$) and $k_r$ a constant specific of each isotope (decay rate) that can be calculated from the radioactive element half life $t_{1/2}$ as $k_r = \ln(2)/t_{1/2}$. Values of $t_{1/2}$ and $k_r$ for common radionuclides are reported





in Table 4. Note that the decay term is more relevant for isotopes with short half lives, *e.g.* for $^{131}$I, which has $t_{1/2} \simeq 8$ days (Brandt et al., 2002; Leelössya et al., 2018).

The radioactive decay term of the isotope $n$ constitutes a sink $I_n^r$ for the isotope itself. However, the decay of the isotope $m$, father of isotope $n$, constitutes a source $S_n^r$ for the isotope $n$:

$$S_n^r = p_{mn} \, I_m^r \tag{56}$$

where $p_{mn}$ is the the relative probability of decay of the isotope $m$ to the isotope $n$. If the isotope $m$ has only one child $n$, the relative probability of the branch $m \mapsto n$ is $p_{mn} = 1$. Note from Table 4 that $^{134}$Cs, $^{137}$Cs and $^{131}$I, decay to stable isotopes, whereas $^{90}$Sr decays to $^{90}$Y which, in turn, is unstable and decays to the stable isotope $^{90}$Zr. The production rate $S_n^r$ of $^{90}$Y is therefore equivalent to the decay rate of $^{90}$Sr.

In *FALL3D-8.0* the radioactive decay is implemented by firstly transporting the radionuclides for a time step $\Delta t$, and then by evaluating the decay during the same time step. For radionuclides that decay to a stable isotope ($^{134}$Cs, $^{137}$Cs and $^{131}$I) it is considered that after a time step $\Delta t$ the concentration decreases as:

$$c(t + \Delta t) = c(t) \, e^{-k_r \Delta t} \tag{57}$$

wheres for decay to an unstable isotope (Yttrium) the concentration varies as:

$$c_Y(t + \Delta t) = \left[ c_{Sr}(t)(1 - e^{-k_{Sr} \Delta t}) + c_Y(t) \right] e^{-k_Y \Delta t} \tag{58}$$

where $c_Y$ and $c_{Sr}$ are, respectively, the concentrations of $^{90}$Y and $^{90}$Sr and $k_Y$ and $k_{Sr}$ are the corresponding decay rates.

### 3.3 Data Insertion

Instrumentation onboard the new generation of geostationary satellites provides with an unprecedented level of spatial resolution and temporal frequency (2 to 4 km pixel size and 10 to 15 min observation period; see Table 5), yielding a quasi-global coverage considering the overlap of different existing platforms. This is very suitable for high-resolution model data assimilation and related uncertainty quantification, as well as to implement ensemble-based dispersal forecast systems. These aspects are still under development and hopefully will be part of next *FALL3D* model distributions. However, *FALL3D-8.0* already includes the possibility of initialising a model run from satellite retrievals. This option, known as data insertion, is typically used in dispersal of volcanic ash and aerosols (mainly $SO_2$) in order to reduce model uncertainties coming from the eruption source term.

Satellite retrievals giving cloud column mass of fine volcanic ash and aerosols can be furnished to the model together with values of cloud thickness, the later needed in order to compute initial concentration (in $\mathrm{kg\,m^{-3}}$) from column mass (in $\mathrm{kg\,m^{-2}}$). In the model initialisation step, gridded satellite data is interpolated into the model grid imposing conservation of mass when concentration values are computed for each model grid cell; *i.e.* ensuring that the resulting column mass in the model (computed concentration times cloud thickness) equals that of satellite data over the same cell area. Examples showing how data insertion improves model accuracy are given in the companion paper (Prata et al., 2019).





## 4 Numerical Implementation

### 4.1 Coordinate mappings and scaling

Consider the ADS equation (1) written in a Cartesian system of coordinates $(x, y, z)$ assuming a diffusion tensor as in (5) and

a sedimentation velocity $\mathbf{u_s} = (0, 0, -w_s)$ aligned with the vertical coordinate $z$:

$$\frac{\partial c}{\partial t} + \frac{\partial (cu)}{\partial x} + \frac{\partial (cv)}{\partial y} + \frac{\partial (cw)}{\partial z} - \frac{\partial (cw_s)}{\partial z}$$

$$-\frac{\partial}{\partial x}\left(k_h \frac{\partial c}{\partial x}\right) - \frac{\partial}{\partial y}\left(k_h \frac{\partial c}{\partial y}\right) - \frac{\partial}{\partial z}\left(k_v \frac{\partial c}{\partial z}\right) = S - I \tag{59}$$

It is straightforward to discretise the above equation in a "brick-like" computational domain $\Omega_c$ using a structured regular (*i.e.* equally-spaced) mesh, although the regularity condition is typically relaxed across the vertical direction so that the vertical grid resolution increases close to ground, where higher gradients are expected. In order to use other coordinate systems, equation

(59) can be written on a generalised orthogonal system of coordinates $(X_1, X_2, X_3)$ (e.g. Toon et al., 1988; Byun and Schere, 2005):

$$\frac{\partial C}{\partial t} + \frac{\partial (CU)}{\partial X_1} + \frac{\partial (CV)}{\partial X_2} + \frac{\partial (CW)}{\partial X_3} - \frac{\partial (CW_s)}{\partial X_3}$$

$$-\frac{\partial}{\partial X_1}\left(K_1 \frac{\partial C}{\partial X_1}\right) - \frac{\partial}{\partial X_2}\left(K_2 \frac{\partial C}{\partial X_2}\right) - \frac{\partial}{\partial X_3}\left(K_3 \frac{\partial C}{\partial X_3}\right) \tag{60}$$

$$= S^* - I^*$$

where $C$ is the scaled concentration, $(U, V, W)$ are the scaled wind components, $(K_1, K_2, K_3)$ are the scaled diffusion coefficients, and $S^*$ and $I^*$ are the scaled source and sink terms. The implementation of a generalised equation like (60) presents

two major advantages. On one hand, the generalised equation reads formally equal to that in Cartesian coordinates, so that little computational penalty exists to map physical domains (*e.g.* accounting for Earth's curvature and topography) to a "brick-like" computational domain (see Figure 1) by using coordinate-dependent horizontal and vertical mappings. On the other hand, a generalised form simplifies the structure and implementation of the code because the model can be solved on various horizontal (cartesian, spherical, Mercator, polar stereographic, etc.) and vertical (terrain following, $\sigma$-coordinates, etc.) coordinate

systems using only one solving routine. To this purpose, one needs first to scale the model coordinates and some terms in the equation using adequate mapping and scaling factors, then solve for the scaled concentration $C$ in the regular computational domain $\Omega_c$ (as in Cartesian coordinates) and, finally, transform the scaled concentration back to the original one.

In general and given two orthogonal coordinate systems $(x_1, x_2, x_3)$ and $(X_1, X_2, X_3)$, coordinate mapping factors are given by the terms $m_{ij}$ of the Jacobian transformation matrix $\mathbf{M}$:

$$m_{ij} = \frac{\partial x_i}{\partial X_j} \tag{61}$$





but, for the transformations considered here, $\mathbf{M}$ will always be diagonal with three non-zero components $m_1$, $m_2$ and $m_3$. For example, in the horizontal transformation to spherical Earth surface coordinates $(\lambda, \phi)$ one has:

$$\begin{aligned} x &= R \, \sin\gamma \, \lambda & &\equiv \sin\gamma \, X_1 \\ y &= R \, \phi & &\equiv X_2 \end{aligned} \tag{62}$$

where $R$ is the radius of the Earth and $\lambda$, $\phi$ and $\gamma$ are the longitude, latitude and colatitude respectively (in Rad). Trivially, this transformation yields to $m_1 = \sin\gamma$ and $m_2 = 1$. Table 6 gives horizontal mapping factors for different coordinate systems (Toon et al., 1988). In most practical cases, *FALL3D-8.0* simulations only consider spherical coordinates, but other options are in principle possible. For vertical transformations, *FALL3D-8.0* incorporates a new $\sigma$-coordinate system with linear decay (Gal-Chen and Somerville, 1975) in which:

$$z = \frac{H-h}{H} X_3 + h \qquad x_3 \in [0, H] \tag{63}$$

where $h(x,y)$ is the terrain height and $H$ the height of the top of the computational domain. In the $\sigma$-coordinate system the influence of the terrain decreases linearly with height, from terrain-following at the surface ($X_3 = 0$) to a rigid lid at the top of the computational domain ($X_3 = H$). This option has been added to partially correct numerical oscillations in the previous terrain-following model mapping ($z = X_3 + h$), which can appear near the surface in case of flows over mountain ranges and propagate upwards (Schar et al., 2002).

Table 7 gives the vertical mapping factors for the different coordinate systems available in *FALL3D-8.0*. Once defined, these coordinate mapping factors are used to scale the variables and parameters that appear in the generalised equation (60). The scaling of scalar quantities is straightforward since it only involves the Jacobian determinant of the transformation, *e.g.* $C = | \mathbf{M} | c = m_1 m_2 m_3 c$ for concentration and so on. Note that for a volume one has $dV = m_1 m_2 m_3 dv$, so that the mass comprised in a cell $dX_1 dX_2 dX_3$ of the computational domain is equal to that in the transformed cell of the physical domain. The horizontal velocity components are trivially scaled as:

$$\begin{cases} U = \dfrac{dX_1}{dt} = u\dfrac{\partial X_1}{\partial x} + v\dfrac{\partial X_1}{\partial y} + w\dfrac{\partial X_1}{\partial z} = u/m_1 \\ V = \dfrac{dX_2}{dt} = u\dfrac{\partial X_2}{\partial x} + v\dfrac{\partial X_2}{\partial y} + w\dfrac{\partial X_2}{\partial z} = v/m_2 \end{cases} \tag{64}$$

whereas for the vertical component one has to consider that, in terrain following coordinate systems, the coordinate $X_3$ depends also on $(x,y)$ through the terrain elevation $h(x,y)$:

$$W = \frac{dX_3}{dt} = u\frac{\partial X_3}{\partial x} + v\frac{\partial X_3}{\partial y} + w\frac{\partial X_3}{\partial z} \tag{65}$$

$$= u\frac{\partial h}{\partial x}\frac{\partial X_3}{\partial h} + v\frac{\partial h}{\partial y}\frac{\partial X_3}{\partial h} + w/m_3 \tag{66}$$

Basic algebra manipulation yields to the scaling factors shown in Table 8 for the different vertical coordinate systems. Note that the expression above implicitly contains the correction for topography in the vertical velocity. Finally, scaling factors for diffusion coefficients can also be obtained after some manipulation (Toon et al., 1988; Byun and Schere, 2006).





## 4.2 Discretisation and solving algorithm

*FALL3D* solves for each model bin the 3D generalised equation (60) using a fractional step method, which splits the equation along each spatial direction (Folch et al., 2009):

$$\frac{\partial C}{\partial t} = S^* - I^*$$

$$\frac{\partial C}{\partial t} + \frac{\partial (CU)}{\partial X_1} - \frac{\partial}{\partial X_1}\left(K_1 \frac{\partial C}{\partial X_1}\right) = 0$$

$$\frac{\partial C}{\partial t} + \frac{\partial (CV)}{\partial X_2} - \frac{\partial}{\partial X_2}\left(K_2 \frac{\partial C}{\partial X_2}\right) = 0 \tag{67}$$

$$\frac{\partial C}{\partial t} + \frac{\partial [C(W - W_s)]}{\partial X_3} - \frac{\partial}{\partial X_3}\left(K_3 \frac{\partial C}{\partial X_3}\right) = 0$$

with the solving order of each resulting one-dimensional ADS equation being permuted in each successive time step in order to avoid any privileged direction. One advantage of this splitting strategy is that, even if each one-dimensional equation is solved

explicitly in time, the result is a semi-implicit scheme that adds stability.

In *FALL3D-8.0*, the "brick-like" computational domain is discretised using a variation of the staggered Arakawa D-grid, in which the wind velocity components are evaluated at the respective cell faces and the rest of scalar quantities at the cell centres (Fig. 2). Note that this configuration is very convenient for solving the 3D equation in a fractional manner because, when solving each one-dimensional case, wind velocities are already aligned with the boundaries of the corresponding one-

dimensional cells. The previous versions of *FALL3D* used the classical Lax-Wendorff (LW) central differences scheme for solving the resulting one-dimensional ADS equations in (67), combined with a slope-limiter to reduce numerical over/under shootings near discontinuities. This resulted on a second-order accuracy except near sharp concentration gradients where, nonetheless, accuracy remained higher than in a first order upwind method. The main advantage of using the LW scheme was its simplicity but, in contrast, it is well known that it introduces numerical dissipation that prevents accurate resolution

of discontinuities, leading to over-diffusive results. In order to circumvent this drawback, *FALL3D-8.0* uses instead a high-resolution Kurganov-Tadmor (KT) scheme that can be combined either with a fourth-order explicit Runge-Kutta or with a first-order Euler time-marching method. The later option, even if less accurate, is still supported for computational efficiency reasons because it implies 4 times less solver calculations.

Consider the general one-dimensional advection-diffusion equation for a scalar variable $c(x,t)$ on its conservative form:

$$\frac{\partial c}{\partial t} + \frac{\partial}{\partial x} F(c) = \frac{\partial}{\partial x} G\left(c, \frac{\partial c}{\partial x}\right) \tag{68}$$

where, in our particular case, $F = c\,u$ is the advective flux and $G = k\partial c/\partial x$ is the diffusive flux (as already introduced, $u(x,t)$ and $k(x,t)$ are the velocity and diffusivity respectively). Consider also a 1D computational domain discretised as in Figure 2, where $c$ is computed at cell centres ("mass" points) and $u$ is stored at staggered cell boundaries, which do not need to be equally spaced. The semi-discrete form of the KT scheme can be written at centre of each cell $i$ in terms of fluxes at boundaries $i \pm 1/2$





as (Kurganov and Tadmor, 2000):

$$
\frac{\partial c_i}{\partial t} = -\frac{1}{\Delta \overline{x}_i}\left[ F^*_{i+1/2} - F^*_{i-1/2}\right] + \frac{1}{\Delta \overline{x}_i}\left[ G^*_{i+1/2} - G^*_{i-1/2}\right]
$$
$$
= f(t,c) \tag{69}
$$

where $\Delta \overline{x}_i$ is the $i$-th cell width and:

$$
F^*_{i+1/2} = \frac{1}{2}\left[ F\left(c^r_{i+1/2}\right) + F\left(c^l_{i+1/2}\right)\right]
$$
$$
-\frac{1}{2}a_{i+1/2}\left(c^r_{i+1/2} - c^l_{i+1/2}\right)
$$
$$
F^*_{i-1/2} = \frac{1}{2}\left[ F\left(c^r_{i-1/2}\right) + F\left(c^l_{i-1/2}\right)\right]
$$
$$
-\frac{1}{2}a_{i-1/2}\left(c^r_{i-1/2} - c^l_{i-1/2}\right) \tag{70}
$$

$$
G^*_{i+1/2} = \frac{1}{2}\left[ G\left(c_i, \frac{c_{i+1}-c_i}{\Delta x_i}\right) + G\left(c_{i+1}, \frac{c_{i+1}-c_i}{\Delta x_i}\right)\right]
$$
$$
= G\left(\frac{c_{i+1}-c_i}{\Delta x_i}\right)
$$
$$
G^*_{i-1/2} = \frac{1}{2}\left[ G\left(c_{i-1}, \frac{c_i-c_{i-1}}{\Delta x_{i-1}}\right) + G\left(c_i, \frac{c_i-c_{i-1}}{\Delta x_{i-1}}\right)\right]
$$
$$
= G\left(\frac{c_i-c_{i-1}}{\Delta x_{i-1}}\right) \tag{71}
$$

Note that, in the expression above, the last equality holds because our flux $G$ depends only on the gradient of $c$, *i.e.* $G = G(\partial c/\partial x)$. In (70), $a_{i\pm 1/2}$ is the maximum absolute value of the eigenvalue of the Jacobian matrix of $F$ (in our particular case,
it reduces to $a_{i\pm 1/2} = |u_{i\pm 1/2}|$), and $c^r$ and $c^l$ are, respectively, right and left values at a cell boundary, computed as:

$$
c^r_{i+1/2} = c_{i+1} - 0.5\phi(r_{i+1})\left(c_{i+1} - c_i\right) \tag{72}
$$
$$
c^r_{i-1/2} = c_i - 0.5\phi(r_i)\left(c_{i+1} - c_i\right)
$$
$$
c^l_{i+1/2} = c_i + 0.5\phi(r_i)\left(c_i - c_{i-1}\right)
$$
$$
c^l_{i-1/2} = c_{i-1} + 0.5\phi(r_{i-1})\left(c_i - c_{i-1}\right)
$$

where

$$
r_i = \frac{c_i - c_{i-1}}{c_{i+1} - c_i} \tag{73}
$$

and $\phi(r)$ is a flux limiter function (e.g. Sweby, 1984). Options available in *FALL3D-8.0* are the well-known superbee $\phi_s(r)$
and minmod $\phi_m(r)$ (Roe, 1986):

$$
\phi_s(r) = \max(0, \min(1, 2r), \min(2, r))
$$
$$
\phi_m(r) = \max(0, \min(1, r)) \tag{74}
$$





Note that $c$ is needed at two extra mass points in order to evaluate $c^l_{i-1/2}$ and $c^r_{i+1/2}$ at the left/right cell boundaries respectively. In other words, the "stencil" of the KT scheme needs two ghost nodes at the boundaries of the computational domain or a two-point halo for internal domains in case of parallel domain decomposition. Note also that the solving strategy derives from a one-dimensional finite-volume formulation that, in our one-dimensional case, is also in practice equivalent to use linear finite elements.

Time marching from $t^n$ to $t^{n+1} = t^n + \Delta t$ in (69) can be performed with the explicit first-order in time Euler method (EU1):

$$c^{n+1} = c^n + \Delta t f\left(t^n, c^n\right) \tag{75}$$

or, alternatively, using the classical fourth-order Runge-Kutta method (RK4), in which:

$$c^{n+1} = c^n + \frac{\Delta t}{6}(k_1 + 2k_2 + 2k_3 + k_4) \tag{76}$$

with:

$$k_1 = f\left(t^n, c^n\right) \tag{77}$$

$$k_2 = f\left(t^n + \frac{\Delta t}{2}, c^n + \frac{\Delta t}{2}k_1\right)$$

$$k_3 = f\left(t^n + \frac{\Delta t}{2}, c^n + \frac{\Delta t}{2}k_2\right)$$

$$k_4 = f\left(t^n + \Delta t, c^n + \Delta t k_3\right) \tag{78}$$

where the function $f(t, c)$ is given by the RHS of (69). In any case, the Courant-Friedrichs-Lewy (CFL) condition can be imposed to guarantee convergence in time integration (e.g. Hindmarsh et al., 1984) along each one-dimensional problem:

$$\Delta t \leq min\left(\frac{1}{\frac{2k_x}{\Delta x^2} + \frac{u}{\Delta y}}, \frac{1}{\frac{2k_y}{\Delta y^2} + \frac{v}{\Delta y}}, \frac{1}{\frac{2k_z}{\Delta z^2} + \frac{w}{\Delta z}}\right) \tag{79}$$

multiplied by a user-defined safety factor. This factor should theoretically be lower than 1 in fully explicit cases but, given the semi-implicit nature of the splitting algorithm, slightly larger values can also yield to stability.

## 4.3 Algorithm benchmarks

Three benchmark cases serve us to illustrate the gains of the KT+RK4 numerical scheme with respect the former LW+EU1 implemented in the previous versions of the code.

**Example 1** considers the pure advection ($k = 0$) of a step-like discontinuity. Consider a 1D domain $x \in [-1, 1]$ with an initial concentration of:

$$\begin{cases} c(t = 0) = 1 & |x| \leq 0.5 \\ c(t = 0) = 0 & |x| > 0.5 \end{cases} \tag{80}$$





that is advected by a uniform velocity field $u = 1$. Periodic conditions are imposed at the boundaries, so that "mass" leaving the computational domain at $x = 1$ is re-injected at $x = -1$. As a result, the initial condition is periodically recovered after each cycle with a period $t = 2$. Results are shown in Figure 3. Note how, as opposed to LW, the KT scheme adds almost no numerical diffusion and preserves discontinuities. In addition, because of its higher order in time, the KT+RK4 scheme

preserves the solution whereas for LW+EU1 accuracy deteriorates with time (compare the accuracy of the LW+EU1 solutions after 1 and 10 cycles in Figure 3).

**Example 2** considers the classical 1D advection-diffusion problem for the onset of numerical instability in a domain $x \in [-1, 1]$ subject to the boundary conditions $c = 0$ at $x = -1$ and $c = 1$ at $x = 1$. The problem has a steady-state analytic solution given by:

$$c(t \to \infty) = \left( e^{Pe(x+1)} - 1 \right) / \left( e^{2Pe} - 1 \right) \tag{81}$$

where $Pe = ul/2k = u/k$ is the Péclet number. Figure 4 shows the steady-state solutions for different Péclet numbers, illustrating also how KT+RK4 outperforms LW+EU1 as the Péclet number increases.

**Example 3** considers a case with pure advection ($k = 0$) on a 2D domain $(x, y) \in [-1, 1] \times [-1, 1]$ with an initial condition at $t = 0$ given by a conic concentration distribution with a unit ($c = 1$) peak concentration centred at $(x_c, y_c) = (0, 0.695)$ and

having a radius $r = 0.1$ (Figure 5a). The cone is advected by a rotating clock-wise velocity field centered at $(0,0)$:

$$u = \Omega\, y$$
$$v = -\Omega\, x \tag{82}$$

with an angular velocity $\Omega = \pi$ (in $\mathrm{Rad\,s^{-1}}$), so that each cycle is repeated with a period of $t = 2$. Concentration profiles along two transects A ($y = y_c$) and B ($x = 0$) after 2 cycles (*i.e.* at $t = 4$) are shown in Figure 5b and 5c respectively. Note again the

substantial improvement in the KT+RK4 scheme, which performs well even in this numerically challenging test.

## 5    Model execution workflow

In *FALL3D-8.0*, the pre-process auxiliary programs have been parallelised and embedded in the code, so that a single executable file exists for all the pre-process steps and execution workflow (see Figure 6). These formerly independent programs can still be run individually as model tasks (specified by a program call argument) or, alternatively, concatenated with the model in a

single execution. In the first case, pre-process tasks generate output files that are later given as inputs to the model task. This is similar to what occurred in the previous v7.x but with the difference of a parallel pre-process. In contrast, the second option does not require intermediate file writing/reading and, therefore, saves disk space and overall computing time. In any case, all tasks share a unique model input file and generate its own log file to track execution and report eventual warnings and errors. Possible task options are summarised in Table 9 and include:

1. Task *SetTgsd*. This task can generate Gaussian and Bi-Gaussian particle grain size distributions in $\Phi$ (log-normal in diameter) or, alternatively, Weibull and Bi-Weibull distributions (Costa et al., 2016a), and assumes a linear variation of





density and particle shape factor between two specified cut-offs. For other kind of grain size distributions, the user must provide a total grain size distribution file (name.tgsd).

2. Task *SetDbs*. This task interpolates all the required meteorological data from the original grid of the driving meteo-
rological models to the computational domain. Table 10 summarises the different meteorological drivers available in *FALL3D-8.0*. Global datasets include ERA-5, the new reanalysis from the ECMWF (Hersbach and Dee, 2016), the NCEP Global Forecast System (GFS), and Global Data Assimilation System (GDAS) final analysis. Regional models supported include the Advanced Research WRF (ARW) core of the Weather Research and Forecasting (WRF) model (Skamarock et al., 2008), the mesoscale models HARMONIE-AROME (Bengtsson et al., 2017) and the COSMO-LAMI,
run by the Italian Regional Environmental Protection Agency (ARPA). Note that some former datasets options have been deprecated in v8.x. The *FALL3D-8.0* distribution package provides a set of utilities to download and pre-process meteorological data for the *SetDbs* model task. Python scripts are provided to download and crop the variables required by the model. ERA-5 can be obtained on either model levels (137 vertical levels) or pressure levels (37 vertical levels) via the Climate Data Store (CDS) infrastructure. GFS datasets can be accessed using the online archive of real-time
weather model output from the National Operational Model Archive and Distribution System (NOMADS) (Rutledge et al., 2006). The NCEP FNL (Final) operational global analysis and forecast data from the Global Data Assimilation System (GDAS) can be accessed using the OPeNDAP protocol through the THREDDS Data Server (TDS) offered by the NCAR Research Data Archive (RDA).

3. Task *SetSrc*. This task generates different emission source terms (see Sec. 3.2.3 and Table 2), including the *PLUME*
option based on the *FPLUME-1.0* model (Folch et al., 2016). If necessary, it also performs a-priori tephra particle aggregation (Sec 3.2.6) and a TGSD cut-off in order to select the effective bins considered in the atmospheric transport.

4. Task *FALL3D*. This task runs the *FALL3D-8.0* model itself.

5. Task *All*. Finally, this task runs all previous tasks consecutively as a single parallel execution.

## 6   Parallelisation and performance

Parallelisation in *FALL3D-8.0* considers a 3D domain decomposition, with freedom for user to choose the number of processors along each direction. In contrast, previous versions considered two different levels of parallelisation, one on particle bins and another on domain but only along the vertical dimension. Parallelisation on bins was convenient in v7.x because no interaction among bins existed, but such a form of trivial parallelism has been deprecated given that it would now yield to unnecessary communication penalties. Note also that the full 3D domain decomposition allows solving on much larger grid sizes before
reaching hardware memory limits.

In terms of run time performance, it is important to recall that the splitting algorithm combined with the RK4 time marching implies solving 4 times a series 1D equations along each spatial dimension, contrasting with the 1 single solution for the EU1



case. In other words, each time integration step of (67) using the RK4 scheme along $X_1$ implies solving 4 times $n_y \times n_z$ one-dimensional problems, the solution along $X_2$ implies solving 4 times $n_x \times n_z$ problems and so on. In order to minimise
the number of communications between neighboring processors, it is more convenient to compute first each of the 4 partial increments in (78) for all the 1D problems on a given dimension. This optimises swapping communications among domain partitions because a single swapping request (*i.e.* one MPI send/receive call) communicates all the data necessary to compute the next RK4 increment for the whole mesh. In contrast, if all partial increments in (78) were computed for each 1D problem, it would require as many swapping requests as 1D problems involved. Clearly, this solving approach is convenient but it presents
two drawbacks. Firstly, it obviously increases the amount of memory required and, secondly, it poses an issue on memory data access, with potential increase of cache memory misses given the more frequent access to data. Moreover, when one solves for the dimensions stored on array lowest axes, memory data access is scattered and therefore less efficient. This issue has been addressed in *FALL3D-8.0* by re-arranging the components of the velocity and diffusion arrays, with the dimension to solve being stored always in the fastest axis, and also by transposing the concentration array that has to be updated at the first partial
increment. Such a memory management strategy allows exploiting always contiguous cache memory positions on any spatial dimension.

All the aspects discussed above have improved substantially the performance and scalability of the code. As an example of strong scaling (*i.e.* time to solution for a fixed problem size), let us consider a real-case ash dispersal simulation from the 2011 Cordón Caulle eruption (e.g. Collini et al., 2013). The model was configured as shown in Table 11 and solved on a typical grid
size of $500 \times 500 \times 100$ computational cells (horizontal model resolution of $\approx 0.03^o$). For illustrative purposes, Figure 7 shows model snapshots of ash cloud column mass at two different times. Figure 8a shows strong scaling results (speed up) up to 2.048 processors obtained on the MareNostrum-IV supercomputer, composed by general-purpose nodes with 48 Intel Xeon Platinum processors interconnected by a 100Gb Intel Omni-Path Full-Fat Tree. For comparison, this Figure shows also the strong scaling curve obtained with latest code version v7.3.4, in this case limited to 64 processors (larger values were not possible on this grid
using v7.3.4 given the former one-dimensional domain decomposition along $z$). Figure 8b plots the corresponding parallel efficiencies, defined as:

$$PE = 100 \times \frac{t_1}{N \, t_N} \tag{83}$$

where $t_1$ is the total computing time with 1 processing unit, and $t_N$ the time with $N$ processing units. Note that ideal strong scaling implies a parallel efficiency of 100%. Clearly, v8.x improves notably the scalability of the code, with a trend close to
that of perfect scaling up to $\approx$100 processors (parallel efficiency $\geq$ 90%). Depending on the time integration scheme, values of parallel efficiency above 50% are obtained with up to 512 and 1024 processors for EU1 and RK4 respectively. This is in striking contrast with version v7.3.4, for which parallel efficiency already drops below 50% with only 16 processors (Fig. 8b). A much better v8.x code performance is also observed not only in terms of code scalability but also in terms of total comput-ing time. Figure 9 shows the computing time ratio (total elapsed time) between v7.3.4 and v.8 depending on the number of
processors and the time integration scheme. The EU1 case (*i.e.* equal order of accuracy in time that in v7.3.4), shows a similar serial performance, only only with an insignificant computation overhead probably due to algorithmic differences. However,



this overhead is rapidly balanced with only 4 processors, and v8.x with EU1 is already up to 4x faster than v7.3.4 with 64 processors. The RK4 case (*i.e.* 4-th order of accuracy in time) is around 4x slower than v.7.3.4 for the serial execution. This is a logical result given that the RK4 scheme needs to iterate 4 times more to solve in time. However, this penalty is also rapidly

balanced as the number of processors increases, and RK4 already outperforms v7.3.4 in absolute terms with only a few teens of processors. In summary, v8.x clearly shows a much better scalability and performance (absolute computing time to solution for a given problem and computational resources) than v7.3.4. The companion paper (Prata et al., 2019) contains a detailed model validation and shows that this is also true in terms of model accuracy.

## 7   Conclusions

After 15+ years, the atmospheric transport model *FALL3D* has been completely rewritten and modernised to overcome legacy constrains in the former release versions v7.x that precluded the introduction of new functionalities and seriously limited the scalability and performance of the code on hundreds or thousands of processors. With this, *FALL3D-8.0* can be considered as a baseline for the successive optimisations and preparation for Exascale computing. However, as detailed in the paper, version v8.x already contains remarkable improvements and updates on model physics, numerics, and performance. In particular, the

code has been prepared to deal also with particles different from tephra, aerosols and radionuclides, includes new coordinate mapping options, a more efficient and less diffusive solving algorithm (KT) that can be combined with a high-order in time solver (RK4), and a better memory management and parallelisation strategy based on a full 3D domain decomposition. Strong scaling results have shown perfect scaling with few hundreds of processors and a parallel efficiency above 50% with 1024 processors. This remarkable improvement is also true in terms of performance (total computing time), with v8.x outperforming

the previous release by a factor of 4x with only 64 processors.

Further expected improvements in the preparation towards Exascale include memory optimisation, introduction of thread parallelism (OpenMP), code vectorisation, porting to accelerators (GPUs), performance portability, load balance, asynchronous I/O, and preparation for emerging heterogeneous architectures (Exascale hardware prototypes).

## 8   Code and data availability

*FALL3D* is available under the version 3 of the GNU General Public License (GPL) at https://gitlab.com/fall3d-distribution

*Author contributions.*  AF and LM have written the bulk of *FALL3D-8.0* with contributions from NG and GM. AC revised and updated all physical parameterisations implemented in the code. NG and MH have performed optimisations and the performance and scalability analysis. AF and AC wrote the manuscript with the input of all coauthors.

*Competing interests.*  The authors declare no competing interests.



*Acknowledgements.* This work has been funded by the H2020 Center of Excellence for Exascale in Solid Earth (ChEESE) under the Grant
Agreement 823844. AC and GM acknowledge the European project EUROVOLC (grant agreement number 731070) and the Ministero
dell'Istruzione, dell'Università e della ricerca (MIUR, Roma, Italy) Ash-RESILIENCE project (grant agreement number 805 FOE 2015).
The authors thank Andrew Prata (BSC) for providing satellite retrievals of ash column mass.





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





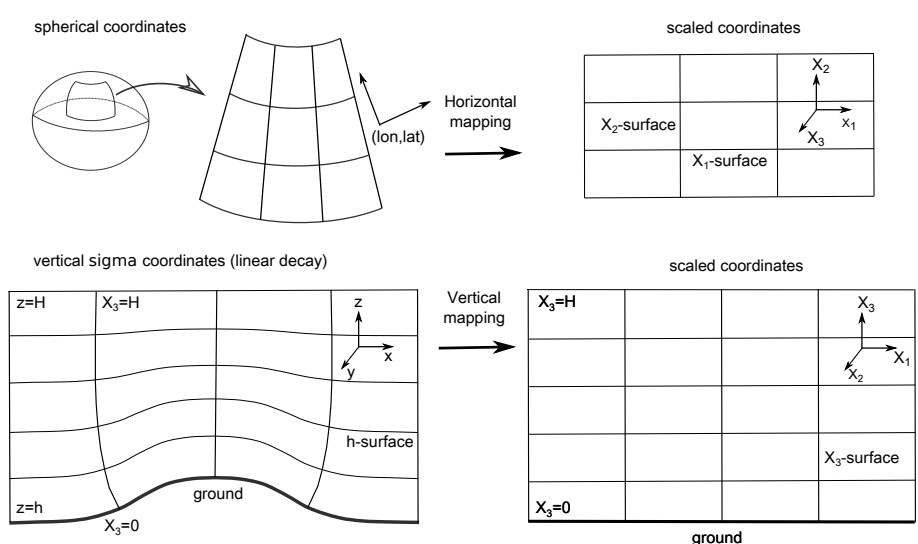

**Figure 1.** Mapping of coordinates from the physical domain (left) to a brick-like (right) computational domain along the horizontal and vertical dimensions. Horizontal mapping corrects for Earth's curvature, vertical mapping accounts for topography.



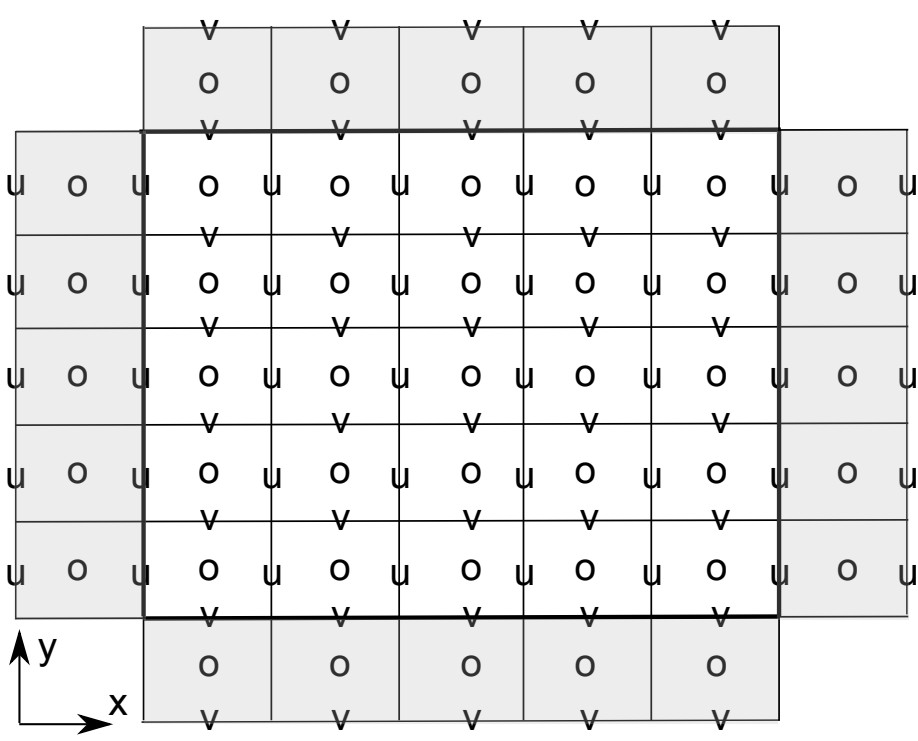

**Figure 2.** Arakawa D-grid on a 2D computational domain limited by the bold line. Scalar quantities are stored/computed at the centre of the cells (empty circles), whereas the $u$ and $v$ velocity components are staggered at its respective cell faces. One row of ghost cells is shown in grey for reference, the actual number of ghost cells needed depends on the numerical stencil (order of the solving algorithm). The 3D grid is formed as a succession of 2D layers, with the $w$ velocity components at the bottom/top faces of the cell.



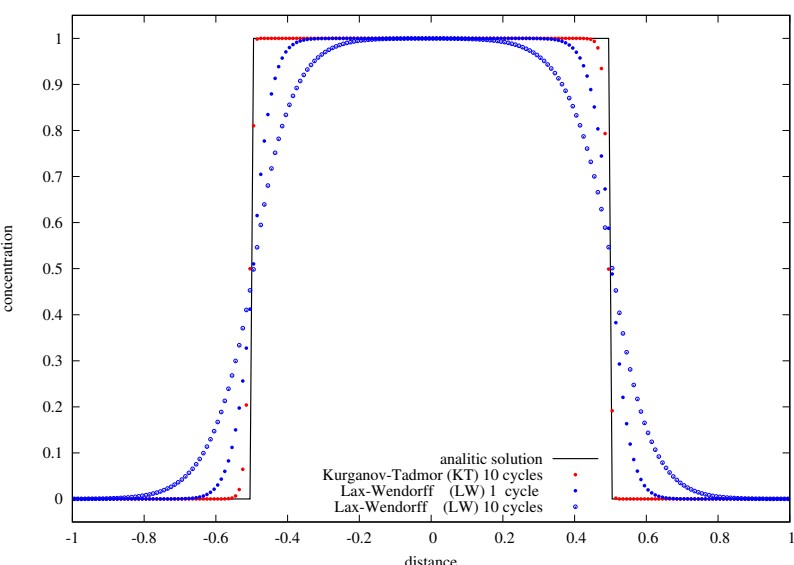

**Figure 3.** Benchmark 1 results. Step-like discontinuity given by (80) moving right at constant velocity $u = 1$ and without diffusion $k = 0$ on a domain $x \in [-1, 1]$ with periodic boundary conditions. The black solid line shows the analytic solution after each periodic cycle, which coincides with the initial condition ($t = 0$). Red dots show the KT+RK4 numerical solution after simulating 10 cycles (at $t = 20$). Blue dots and blue circles show the LW+EU1 results after 1 cycle ($t = 2$) and 10 cycles ($t = 20$) respectively. Note that mass (area below curves) is conserved in all the cases, but the KT+RK4 scheme adds almost no numerical diffusion and preserves discontinuities. Results using 200 equally-spaced grid cells.



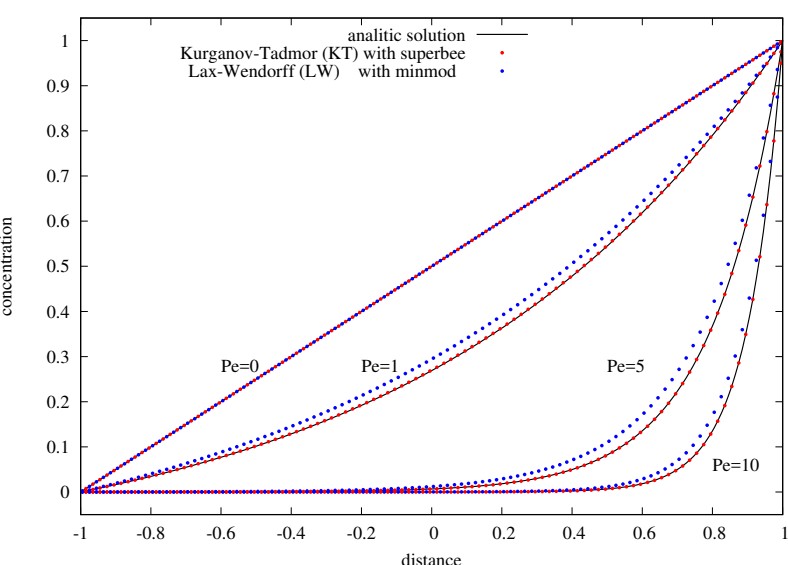

**Figure 4.** Benchmark 2 results. Steady-state solutions for different Péclet numbers ($Pe$) on a domain $x \in [-1, 1]$ with boundary conditions $c = 0$ at $x = -1$ and $c = 1$ at $x = 1$. Black lines show the respective analytic solutions given by (81). Red and blue dots show, respectively, the KT+RK4 and LW+EU1 numerical solutions. The case $Pe = 0$ (pure diffusion problem) has a linear solution and is exactly matched by both schemes. For the rest of cases KT+RK4 outperforms LW+EU1. Results using 200 equally-spaced grid cells.

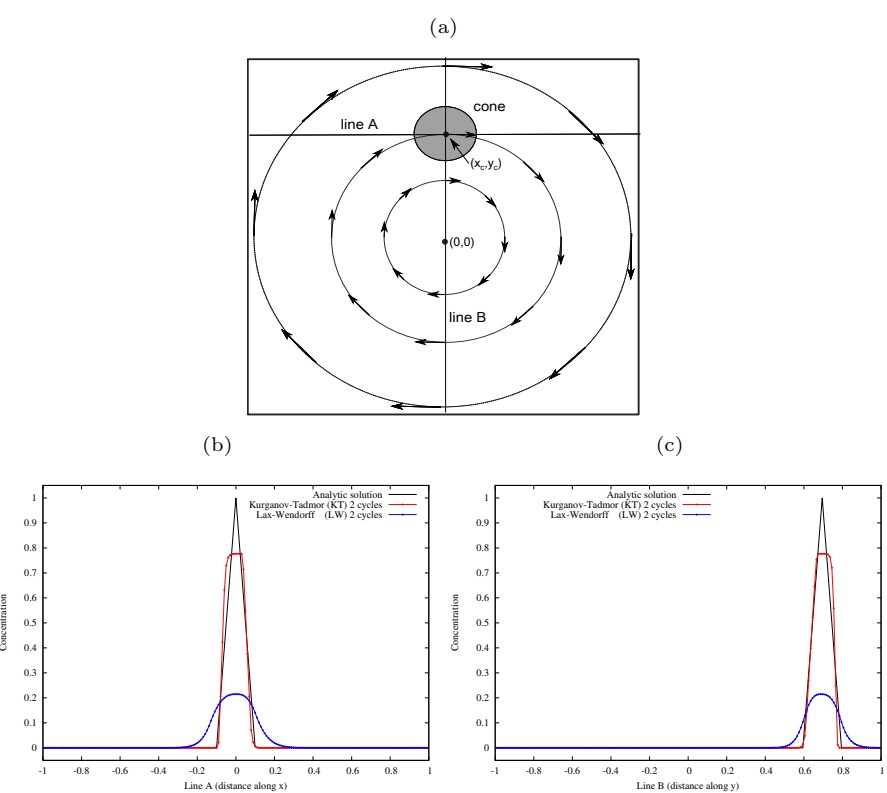

**Figure 5.** Benchmark 3 results. (a) A cone centred at $(x_c, y_c) = (0, 0.695)$ with radius $r = 0.1$ (shaded circle) is advected by a rotating velocity field with angular velocity $\Omega = \pi$ in a computational domain $(x, y) \in [-1, 1] \times [-1, 1]$. Plots (b) and (c) show concentration profiles after 2 cycles along lines A ($y = y_c$) and B ($x = 0$). The black solid line shows the analytic solution after each cycle. Red and blue dots show the KT+RK4 and LW+EU1 numerical solutions respectively. Results using 200 equally-spaced cells along each direction



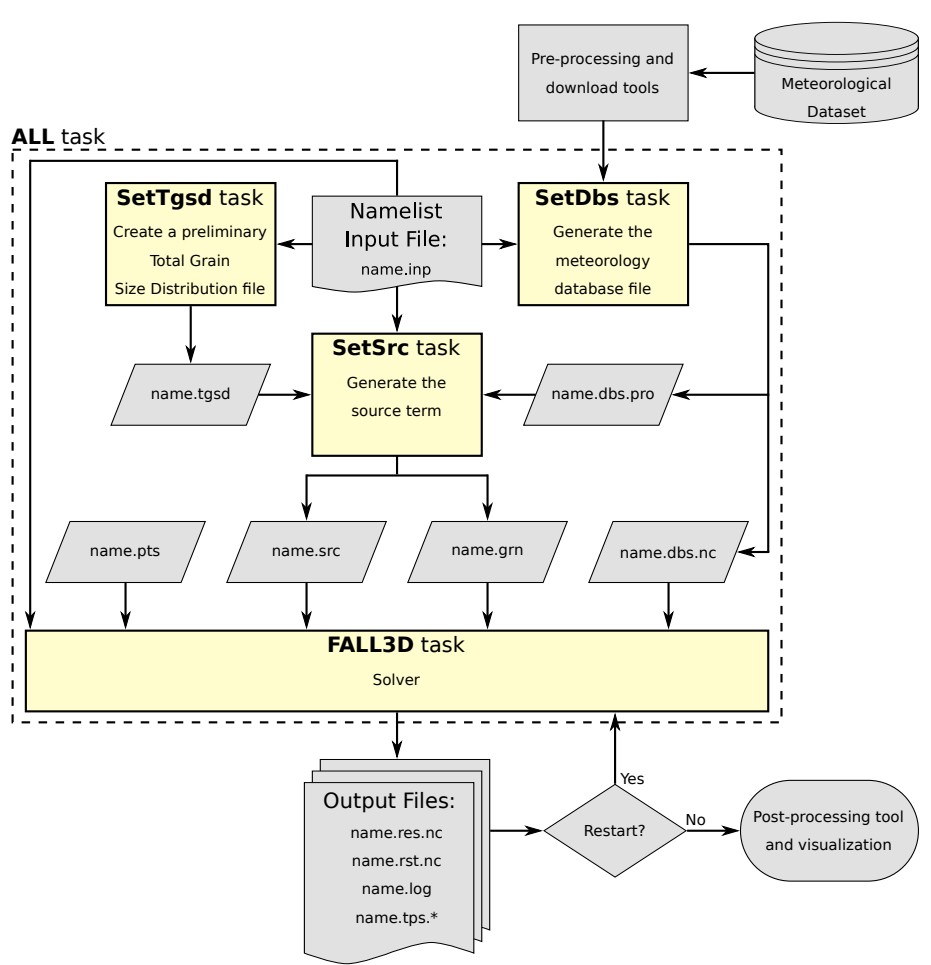

**Figure 6.** Pre-process and execution workflow and associated model tasks. Task "All" runs all tasks in a single (parallel) execution. In this case, the write/read of the intermediate files can be omitted. All tasks share a unique model input file (name.inp).



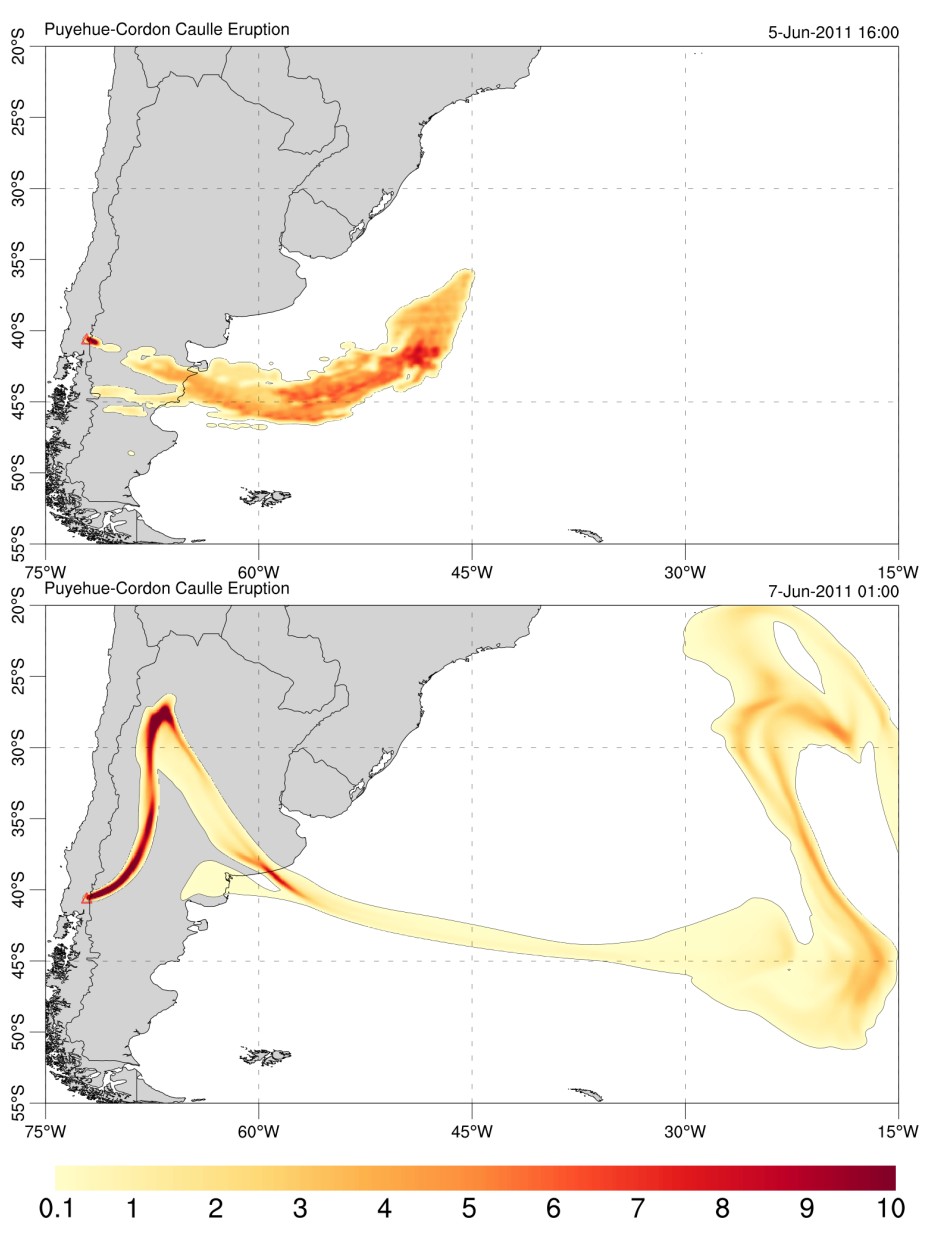

**Figure 7.** Simulation results for the 2011 Cordón Caulle eruption. The model was configured as shown in Table 11 and solved on the $500 \times 500 \times 100$ grid cells used in Section 6 for the scalability analysis. (a) PM10 ash cloud column mass contours (in $\mathrm{kg\,m^{-2}}$) on 5 June 2011 at 16:00 UTC, just after satellite data insertion. (b) same on 7 June 2011 at 01:00 UTC after model evolution assuming a constant column height of 10 km a.v.l.





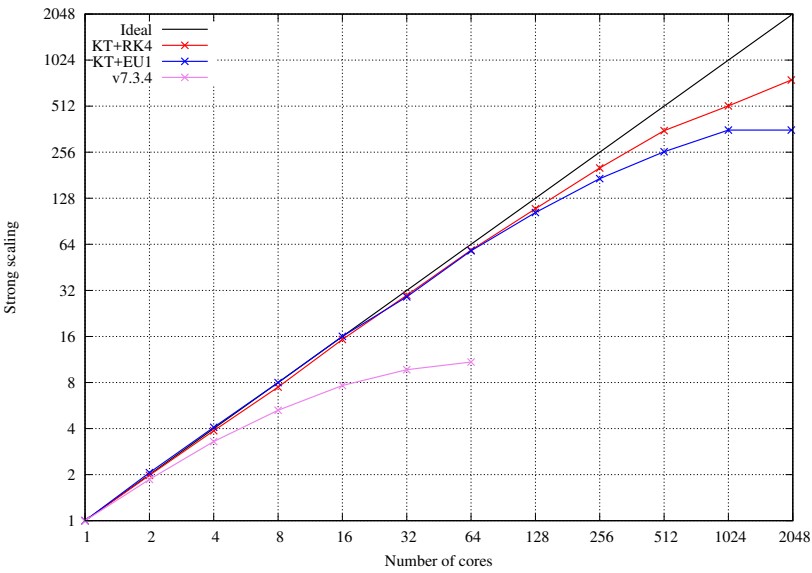

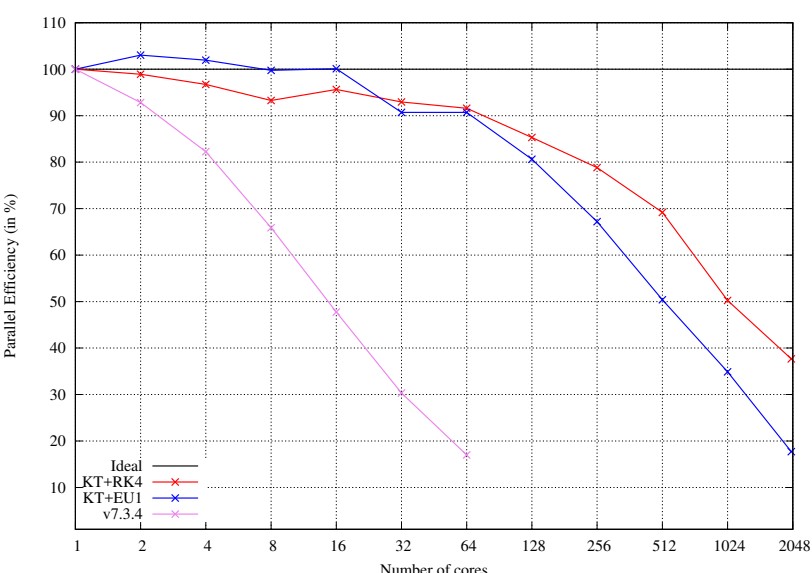

**Figure 8.** Strong scaling results on MareNostrum-IV supercomputer for the Cordón Caulle simulation using a $500 \times 500 \times 100$ cells grid. (a) scaling curves up to 2048 processors for the RK4 (red) and EU1 (blue) time integration schemes. Results using code version v.7.3.4 are also shown up to 64 processors only (pink). Ideal strong scaling behaviour is indicated by the solid black line. Note that the scaling curves refer to the total computing time and therefore include I/O operations, not just computing time. (b) Parallel efficiency (83) depending on the number of computation units.



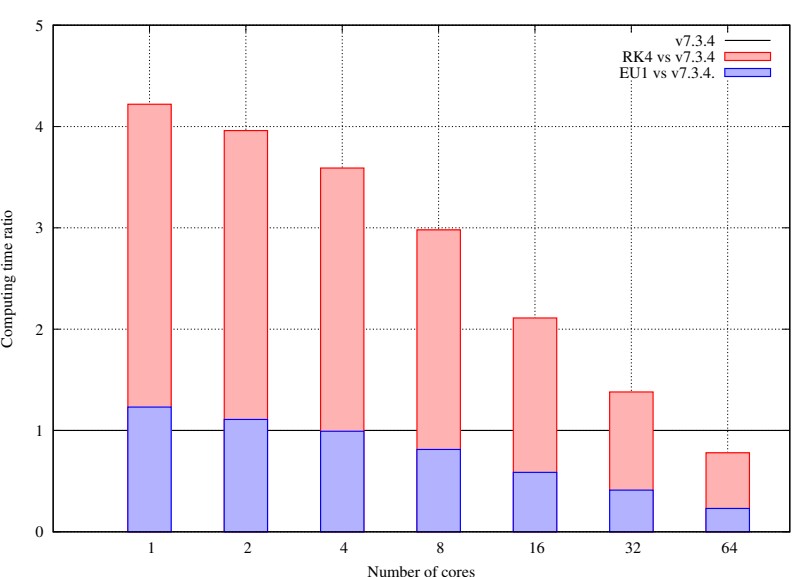

**Figure 9.** Computing time ratio between code version v7.3.4 and v.8 depending on the number of processors. Results for EU1 (blue bars) and RK4 (red bars) including I/O operations.

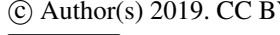



**Table 1.** Types of categories and related sub-categories of species in *FALL3D-8.0*.

| category | sub-category (species) | name (tag) | bins (number) | comments |
|---|---|---|---|---|
| particles | tephra | lapilli | user defined[1] | tephra with $\Phi$[2] $< -1$ |
| | | coarse-ash | user defined | tephra with $-1 \leq \Phi \leq 4$ |
| | | fine-ash | user defined | tephra with $\Phi > 4$ |
| | | aggregate | 1 or more | aggregation model dependent |
| | mineral dust | dust | user defined[1] | |
| aerosols | $H_2O$ | H2O | 1 | water vapour |
| | $SO_2$ | SO2 | 1 | sulphur dioxide only[3] |
| radionuclides | $^{134}$Cs | CS-134 | user defined[1] | cesium 134, decays to a stable isotope |
| | $^{137}$Cs | CS-137 | user defined | cesium 137, decays to a stable isotope |
| | $^{131}$I | I-131 | user defined | iodium 131, decays to a stable isotope |
| | $^{90}$Sr | SR-90 | user defined | stroncium 90, decays to yttrium 90 |
| | $^{90}$Y | Y-90 | user defined | yttrium 90, decays to a stable isotope |

(1) For any specie in the category particles or radionuclides, users can specify the number of effective bins from a grain size distribution;

(2) for tephra, the $\Phi$ number is defined as $d = 2^{-\Phi}$, where $d$ is the particle diameter in $\mathrm{mm}$; (3) $SO_2$ chemistry not included yet in v8.0.





**Table 2.** Parameterisations available in *FALL3D-8.0* depending on each category and related sub-category (species). Note that, except for radioactive decay, all physical phenomena were already included in v7.x. However, several parameterisations have been updated to account for more recent developments.

| Category<br>Sub-category (species) | | particles | | aerosols | radionuclides |
|---|---|---|---|---|---|
| | | tephra | dust | (all species) | (all species) |
| Diffusion (Sec. 3.2.1) | | √ | √ | √ | √ |
| Particle sedimentation (Sec. 3.2.2) | | √ | √ | | √ |
| Emissions (Sec. 3.2.3) | *POINT* | √ | | √ | √ |
| | *HAT* | √ | | √ | √ |
| | *SUZUKI* | √ | | √[1] | √ |
| | *PLUME* | √ | | √[1] | |
| | *RESUSPENSION* | √ | √ | | |
| Deposition mechanisms (Sec. 3.2.4) | dry deposition [2] | √ | √ | √ | √ |
| | wet deposition [3] | √ | √ | √ | √ |
| Gravity current (Sec. 3.2.5) | | √ | | √[1] | |
| Aggregation (Sec. 3.2.6) | | √ | | | |
| Radiaoctive decay (Sec. 3.2.7) | | | | | √ |
| Chemical reactions | | | | √[4] | |

(1) only for volcanic aerosols, (2) applies only to particles/aerosols smaller than 100 μm, (3) cut-off at 100 and 1 μm assumed for below and in-cloud scavenging respectively, (4) not included yet in v8.0.



**Table 3.** Options available in *FALL3D-8.0* for vertical distribution of mass and total source strength depending on the type of emission source $S^e$.

| Type of source | Vertical distribution of mass | Total source strength $M_o$ options | Comments |
|---|---|---|---|
| *POINT* *HAT* *SUZUKI* | Prescribed (same for all bins) | (i) Prescribed; (ii) Given by (29); (iii) Given by (30) | Vertical distributions valid for any type of bin but source strength options (ii) and (iii) are for volcanic particles only |
| *PLUME* | Computed by the *FPLUME-1.0* model (bin dependent) | (i) Prescribed; (ii) computed by *FPLUME-1.0* model from eruption column height (inverse problem) | Only for volcanic particles/aerosols |
| *RESUSPENSION* | Distributed linearly within the ABL or assigned to the first vertical model layer (bin dependent) | Computed from surface cell area and vertical flux using emission schemes (33), (34), or (36) | Only for resuspended particles (ash and dust) |

**Table 4.** List of radionuclides implemented in the model. Table shows half life, decay rate (in $\mathrm{s}^{-1}$) and resulting child product.

| Radionuclide | $t_{1/2}$ | $t_{1/2}$ (s) | $k_r$ ($\mathrm{s}^{-1}$) | Product |
|---|---|---|---|---|
| $^{134}$Cs | 2.065 years | $6.51 \times 10^7$ | $1.06 \times 10^{-8}$ | $^{134}$Ba (stable) |
| $^{137}$Cs | 30.17 years | $9.51 \times 10^8$ | $7.29 \times 10^{-10}$ | $^{137}$Ba (stable) |
| $^{131}$I | 8.0197 days | $6.93 \times 10^5$ | $1.00 \times 10^{-6}$ | $^{131}$Xe (stable) |
| $^{90}$Sr | 28.79 years | $9.08 \times 10^8$ | $7.63 \times 10^{-10}$ | $^{90}$Y (unstable) |
| $^{90}$Y | 2.69 days | $2.33 \times 10^5$ | $2.98 \times 10^{-6}$ | $^{90}$Zr (stable) |

**Table 5.** Characteristics of sensors for ash and $SO_2$ detection onboard new generation of geostationary satellites. Table courtesy from Andrew Prata.

| Satellite | Sensor | Coverage | Spatial res.(km) | Temporal res.(min) | Ash/SO2 bands ($\mu$m) | Lifetime |
|---|---|---|---|---|---|---|
| Meteosat-11 | SEVIRI | Europe and Africa | 3 | 15 | 7.35, 8.7, 10.8, 12 | 2015-2022 |
| FY-4A | AGRI | S. Asia and Oceania | 4 | 15 | 8.5, 10.7, 12 | 2016-2021 |
| Himawari-8 | AHI | S. Asia and Oceania | 2 | 10 | 7.35, 8.6, 10.45, 11.2, 12.35 | 2014-2029 |
| GOES-17 | ABI | W. America | 2 | 10 | 7.4, 8.5, 10.3,11.2, 12.3 | 2018-2029 |
| GOES-16 | ABI | E. America | 2 | 10 | 7.4, 8.5, 10.3,11.2, 12.3 | 2016-2027 |





**Table 6.** Horizontal mapping factors $(x, y) \leftarrow (X_1, X_2)$ for different coordinate systems where $\lambda$ is longitude, $\phi$ latitude, $R$ the radius of the Earth, $\gamma$ colatitude, and $\phi_o$ the latitude at which the projection is true. Mercator and polar stereographic projections use cartesian coordinates projected into the Earth surface, *i.e.* account for curvature. Regular Cartesian coordinates should be used only for local domains, where the Earth's curvature can be neglected.

| Coordinate system | | | $m_1$ | $m_2$ |
|---|---|---|---|---|
| Regular $(x, y)$ | $X_1 = x$ | $X_2 = y$ | 1 | 1 |
| Mercator $(x, y)$ | $X_1 = x$ | $X_2 = y$ | $\dfrac{\cos\phi}{\cos\phi_o}$ | $m_1$ |
| Polar stereographic $(x, y)$ | $X_1 = x$ | $X_2 = y$ | $\dfrac{1 + \sin\phi}{1 + \sin\phi_o}$ | $m_1$ |
| Spherical $(\lambda, \phi)$ | $X_1 = R\,\lambda$ | $X_2 = R\,\phi$ | $\sin\gamma$ | 1 |

**Table 7.** Vertical mapping factors $z \leftarrow X_3$ for different coordinate systems where $h(x, y)$ is topography and $H$ the top of the computational domain.

| Coordinate system | | $m_3$ |
|---|---|---|
| Regular | $X_3 = z$ | 1 |
| Full terrain following | $X_3 = z - h$ | 1 |
| $\sigma$ linear decay | $X_3 = \dfrac{z - h}{H - h} H$ | $\dfrac{H - h}{H}$ |



**Table 8.** Scaling factors for the different terms in the generalised coordinates ADS equation (60).

| Variable/parameter | scaling |
|---|---|
| Horizontal velocities | $U = u/m_1 \qquad V = u/m_2$ |
| Vertical (and settling) velocity | |
| Regular | $W = w/m_3$ |
| Full terrain following | $W = \left[ -u\dfrac{\partial h}{\partial x} - v\dfrac{\partial h}{\partial y} + w \right]/m_3$ |
| $\sigma$ linear decay | $W = \left[ -u\left(1 - \dfrac{X_3}{H}\right)\dfrac{\partial h}{\partial x} - v\left(1 - \dfrac{X_3}{H}\right)\dfrac{\partial h}{\partial y} + w \right]/m_3$ |
| Horizontal diffusion coefficients | $K_1 = k_h/m_1^2 \qquad K_2 = k_h/m_2^2$ |
| Vertical diffusion coefficient | |
| Regular | $K_3 = k_v/m_3^2$ |
| Full terrain following | $K_3 = \left[ k_h\left(\dfrac{\partial h}{\partial x}\right)^2 + k_h\left(\dfrac{\partial h}{\partial y}\right)^2 + k_v \right]/m_3^2$ |
| $\sigma$ linear decay | $K_3 = \left[ k_h\left(1 - \dfrac{X_3}{H}\right)^2\left(\dfrac{\partial h}{\partial x}\right)^2 + k_h\left(1 - \dfrac{X_3}{H}\right)^2\left(\dfrac{\partial h}{\partial y}\right)^2 + k_v \right]/m_3^2$ |
| Concentration and source/sink | $C = m_1 m_2 m_3\, c \qquad S^* = m_1 m_2 m_3\, S \qquad I^* = m_1 m_2 m_3\, I$ |

**Table 9.** Summary of the *FALL3D-8.0* model tasks.

| Task | Task call *arguments* | Comments |
|---|---|---|
| SetTgsd | Fall3d.x SetTgsd *problemname.inp*[1] | Runs the SetTgsd pre-process utility |
| SetDbs | Fall3d.x SetDbs *problemname.inp [npx npy npz]*[2] | Runs the SetDbs pre-process utility |
| SetSrc | Fall3d.x SetSrc *problemname.inp [npx npy npz]* | Runs the SetSrc pre-process utility |
| Fall3d | Fall3d.x Fall3d *problemname.inp npx npy npz* | Runs *FALL3D-8.0* |
| All | Fall3d.x All *problemname.inp npx npy npz* | Runs all previous tasks in a single execution |

(1) model input file (same for all tasks), (2) number of processors along each spatial dimension in domain decomposition. If not given, the execution is serial.





**Table 10.** Summary of the meteorological drivers available in *FALL3D-8.0*.

| ID | Map Projection | H. Res. | Time Res. | Vertical Coord. | Vertical Levels | Period | Format |
|---|---|---|---|---|---|---|---|
| *Global model forecasts* | | | | | | | |
| GFS (NCEP) | Regular lat-lon | 0.25° | 1h | Isobaric | 34 | +384h | GRIB2 |
| | | 0.5° | 3h | | 50 | +384h | |
| | | 1.0° | 3h | | 34 | +384h | |
| *Global model final analyses and reanalyses* | | | | | | | |
| GDAS (NCEP) | Regular lat-lon | 0.25° | 6h | Isobaric | 31 | 2012-present | GRIB2 |
| ERA5 | Regular lat-lon | >0.25° | 1h | Isobaric | 37 | 1979-present | netCDF or GRIB1 |
| ERA5ML | Regular lat-lon | >0.25° | 1h | Hybrid | 137 | 1979-present | netCDF or GRIB2 |
| *Mesoscale models* | | | | | | | |
| WRF | Regular lat-lon Lambert Conformal Mercator Polar stereographic | user-defined | user-defined | Terrain following/ Hybrid | user-defined | user-defined | netCDF |
| HARMONIE-AROME | Lambert Conformal | 2.5 km | 1h | Hybrid | 65 | user-defined | GRIB |
| COSMO-LAMI (ARPA) | Mercator | user-defined | 3h | Isobaric | 14 | user-defined | GRIB |

**Table 11.** *FALL3D-8.0* model configuration for the Cordón Caulle simulation example shown in Section 6

| Variable/parameter | Configuration |
|---|---|
| Computational domain | $500 \times 500 \times 100$ grid cells with top at 15 km a.s.l. |
| Coordinate mappings | spherical and $\sigma$ linear decay |
| Horizontal resolution | $\approx 0.03^o$ |
| Vertical resolution | 150 m (in the mapped computational domain) |
| Run start time | 5 June 2011 at 15:00 UTC |
| Initial condition | Data insertion from GOES at 2 km spatial resolution |
| Driving meteorological data | ERA-5 reanalysis (pressure levels) |
| TGSD | Estimated from column height as in Costa et al. (2016a) |
| Ash bins | 5 effective bins with cut-off at $\Phi = 5$ ($32\mu m$ effective diameter) |
| Ash aggregation | None |
| Column height | 10 km a.v.l. (sustained from data insertion onwards) |
| Emission source | *SUZUKI* option with $A = 4$ and $\lambda = 1$ |
| Terminal velocity model | *GANSER* option |
| Turbulent diffusion | As in Byun and Schere (2006) (horizontal) and similarity theory (vertical) |
| Deposition mechanisms | None |