# Peer review of "FALL3D-8.0: a computational model for atmospheric transport and deposition of particles, aerosols and radionuclides. Part I: model physics and numerics"

_Geoscientific Model Development, 2019_

## Referee Comment (RC1) · Fabio Dioguardi (Referee) · 6 Jan 2020

This manuscript present the new version of FALL3D, a widely used and known dispersion model. The new version of FaLL3D here presented includes substantial improvements, particularly in the numerical method for solving the advection-diffusion-sedimentation equation and the parallel implementation. The former, thanks to the implementation of a less diffusive scheme, will prevent the numerical diffusion that characterized previous version of FALL3D, hence allowing to better capture sharp gradients of the computed field (e.g. ash concentration in the atmosphere). This has a

fundamental importance, e.g., for volcanic ash dispersion simulations, i.e. for better defining the position and extent of the volcanic ash cloud in the atmosphere. The parallel implementation has been substantially improved by the implementation of the domain decomposition method, which will significantly reduce the required computation time and optimize the usage of FALL3D on supercomputers. No need to mention the importance of this improvement in particular in terms of the run-time of dispersion simulations, especially when high-resolution computation with different grainsize bins are going to be used for near real-time forecasts. The manuscript is well organized, written and supported by tables, figures and references. The two reasons detailed above already serve as I have some minor comments I wish the authors to address. These are also visible in the attached highlighted version of the PDF document. Specifically:

- Abstract. I would like the authors to add some more explicit conclusive statements on the impact of the improvements of FALL3D, particularly the implication and possible future applications that are now possible thanks to the new features.

- Line 67-69. Could the authors provide more detail here? To my knowledge, all model parametrizations of the volcanic source (a part from more complex models) assume a relationship between plume height/trajectory and emission rate at the source, regardless the grainsize distribution. Hence, total emission rate should always apply to the whole granulometric spectrum. Why do the authors write "several"? Can they provide examples for which the above does not necessarily apply?

- Line 118. I would like the authors to give more insight on the limitations/consequences of the "passive transport" assumption for solid particles here. Could they explain which is, e.g., the maximum particle size for which this assumption may be considered reasonable?

- Line 160. Could the author give more insight and/or instruction to the reader and model user on the "characteristic grid cell measure"?

- Line 262. Is there a particular reason why the model of Degruyter & Bonadonna

(2012) has been removed from FALL3D?

- Please check the use of symbols throughout the manuscript, some symbols have been used twice for different physical quantities/constant. Some examples are highlighted in the attached manuscript but I urge the authors to review all symbols and possibly add a Symbol list table.

Other few minor corrections are suggested in the attached document.

For all the above I recommend the manuscript to be accepted for publication after minor revisions addressing the points above have been made.

Fabio Dioguardi, PhD

---

## Referee Comment (RC2) · Anonymous Referee #2 · 8 Jan 2020

The manuscript presents the latest version of FALL3D, a well-known and widely used Eularian pollutant dispersal model. The model code has been rewritten from scratch to remove legacy issues and add new features such as a new solving strategy to reduce numerical diffusion, new species (up till version 7.x the code only included tephra and SO2), and a new much-improved parallelisation strategy. For all intents and purposes FALL3D-8.0 is practically a new model and as such I believe merits publication of a new paper.

The manuscript is well-written, introduces the additions and changes to the code so

that users of the previous versions can understand the differences, but also reintroduces critical but unmodified concepts to keep the paper suitable for new users. There are only a few minor changes that I think would benefit the paper (listed below and in the attached PDF), so I would suggest accepting the manuscript after minor revisions.

First, a small point about the naming. As a user of FALL3D 7.x I find that the change from version 7.x to 8.0 as detailed in the manuscript is so significant that, in my opinion, the authors could be presenting a new (but familiar) model (similar to for example the change from MM5 to WRF). In this respect I feel that just changing the version number in a sense undersells the radical changes in the code presented. But I can understand the authors decision in order to keep the familiarity with the model name (although not all of the species fall anymore!).

Aside from some technical comments that are shown in the attached PDF, the main change that I would like the authors to implement would be to add an appendix or a supplemental document highlighting the changes aimed at previous users. For example it would help to add a table listing the removed "deprecated" options (e.g. ln 73, 143), with perhaps a one-sentence explanation of why the option was removed when appropriate. For most of the removed options I can understand the logic, but I feel that it would be a good addition to have these changes summarized at some point.

Another point that I would like to note is that, after downloading the code from gitlab, I found that even though the compiling and execution process has been streamlined considerably, and the namelist file has been reworked to be more user-friendly, it has changed to the extent that previous users require some guidance to adjust to the modified workflow (i.e. the change in the run directories, scripts, etc). I understand that this might be better suited as a subsection in the new version's manual rather than this manuscript, but I would appreciate either summarizing these changes as a supplemental document to the paper, or at least, after the model is properly released and advertised, making sure to add such a section to the new version of the manual.

Overall, this is a very exciting step for the model and for the relevant modelling communities. I would like to wish the writers the best of luck with the revisions and I'm looking forward to using the new iteration of the model.

Minor and technical comments: See manuscript for minor comments (blue) and technical corrections (green).

Please also note the supplement to this comment:
https://www.geosci-model-dev-discuss.net/gmd-2019-311/gmd-2019-311-RC2-supplement.pdf

**Supplement:**

[revised manuscript text omitted]

---

## Author Comment (AC1) · 10 Feb 2020

We thank reviewer#1 (Fabio Dioguardi) for his constructive review.

Q1. Abstract. I would like the authors to add some more explicit conclusive statements on the impact of the improvements of FALL3D, particularly the implication and possible future applications that are now possible thanks to the new features.

R1. We added the following sentence to the abstract: "All these new features and improvements have implications on operational model performance and allow, among

other, adding data assimilation and ensemble forecast in future releases."

Q2. Line 67-69. Could the authors provide more detail here? To my knowledge, all model parametrizations of the volcanic source (a part from more complex models) assume a relationship between plume height/trajectory and emission rate at the source, regardless the grainsize distribution. Hence, total emission rate should always apply to the whole granulometric spectrum. Why do the authors write "several"? Can they provide examples for which the above does not necessarily apply?

R2. Some resuspension schemes (e.g. for tephra or dust emission) give emission rate for each particle bin. This is not the case of volcanic plumes, for which emission schemes are always parameterized in terms of the total grainsize distribution. To avoid confusion we have replaced the word "several" by "volcanic plume source parameterisations".

Q3. Line 118. I would like the authors to give more insight on the limitations/consequences of the "passive transport" assumption for solid particles here. Could they explain which is, e.g., the maximum particle size for which this assumption may be considered reasonable?

R3. The "passive transport" approach assumes that particles do not interact (dilute concentration) and that, except for the settling velocity term, are coupled with the carrier fluid. This means that the particle Stokes number is "low", which in the case of air typically holds up to few millimeters. We added the following sentence: "Note that the passive transport equation (1) neglects inertial terms and, consequently, assumes low particle Stokes number."

Q4. Line 160. Could the author give more insight and/or instruction to the reader and model user on the "characteristic grid cell measure"?

R4. The model use the equivalent area length for eq(8). We added the following sentence: (e.g. the equivalent area length)

[Figure]

Q5. Line 262. Is there a particular reason why the model of Degruyter & Bonadonna (2012) has been removed from FALL3D?

R5. This parameterization gives similar results to that of Woodhouse but, given the structure of meteo data profiles in the code, has a much larger computational penalty and has been removed for this reason.

Q6. Please check the use of symbols throughout the manuscript, some symbols have been used twice for different physical quantities/constant. Some examples are highlighted in the attached manuscript but I urge the authors to review all symbols and possibly add a Symbol list table.

R6. Thank you; the (several) repeated symbols have been corrected. In addition, we added 2 new Tables containing the list of Latin and Greek symbols as suggested by the referee.

---

## Author Comment (AC2) · 10 Feb 2020

We thank reviewer#2 (anonymous) for his constructive review.

Q1. The main change that I would like the authors to implement would be to add an appendix or a supplemental document highlighting the changes aimed at previous users.

R1. This is what (and why) has been deprecated:

- Estimate MER from H using the Degrutyer model: This parameterization gives very similar results to that of Woodhouse but, given the structure of meteo data profiles in the code, has a much larger computational penalty

- RAMS horizontal diffusion: This parameterization has been replaced by that proposed by Byun and Schere (2006), which is similar to RAMS option but preferable as allow for reducing the dependency of horizontal diffusion on grid resolution.

- SURFACE_LAYER vertical diffusion: Parameterizations for describing diffusivity tensor were updated adopting those used in the CAM-4.0 model (Neale et al., 2010)

- Meteo datasets: ERA40, ERA-Interim, NCEP reanalysis 1 and 2 at 2.5: Replaced by more updated equivalent datasets at higher resolution

These are actually few options, probably insufficient for an appendix. Will be added as a note in the model user's manual.

Q2. I found that even though the compiling and execution process has been streamlined considerably, and the namelist file has been reworked to be more user-friendly, it has changed to the extent that previous users require some guidance to adjust to the modified workflow.

R2. We understand the concern. This is now addressed in the model user manual available online: https://gitlab.com/fall3d-distribution/v8.0/-/wikis/home

Q3. Minor and technical comments (attached documnent)

R3. Changes accepted.